# TRPM7 is critical for short-term synaptic depression by regulating synaptic vesicle endocytosis

**Zhong-Jiao Jiang[1†], Wenping Li[1†], Li-Hua Yao[1,2], Badeia Saed[3], Yan Rao[1], Brian S Grewe[1], Andrea McGinley[1], Kelly Varga[1,4], Simon Alford[5], Ying S Hu[3], Liang-Wei Gong[1]\***

[1]Department of Biological Sciences, University of Illinois at Chicago, Chicago, United States; [2]School of Life Science, Jiangxi Science & Technology Normal University, Nanchang, China; [3]Department of Chemistry, University of Illinois at Chicago, Chicago, United States; [4]Department of Biological Sciences, University of North Texas at Dallas, Dallas, United States; [5]Department of Anatomy and Cell Biology, University of Illinois at Chicago, Chicago, United States

**\*For correspondence:**
lwgong@uic.edu

[†]These authors contributed equally to this work

**Competing interest:** The authors declare that no competing interests exist.

**Abstract** Transient receptor potential melastatin 7 (TRPM7) contributes to a variety of physiological and pathological processes in many tissues and cells. With a widespread distribution in the nervous system, TRPM7 is involved in animal behaviors and neuronal death induced by ischemia. However, the physiological role of TRPM7 in central nervous system (CNS) neuron remains unclear. Here, we identify endocytic defects in neuroendocrine cells and neurons from TRPM7 knockout (KO) mice, indicating a role of TRPM7 in synaptic vesicle endocytosis. Our experiments further pinpoint the importance of TRPM7 as an ion channel in synaptic vesicle endocytosis. $Ca^{2+}$ imaging detects a defect in presynaptic $Ca^{2+}$ dynamics in TRPM7 KO neuron, suggesting an importance of $Ca^{2+}$ influx via TRPM7 in synaptic vesicle endocytosis. Moreover, the short-term depression is enhanced in both excitatory and inhibitory synaptic transmissions from TRPM7 KO mice. Taken together, our data suggests that $Ca^{2+}$ influx via TRPM7 may be critical for short-term plasticity of synaptic strength by regulating synaptic vesicle endocytosis in neurons.

## Introduction

Transient receptor potential melastatin 7 (TRPM7), a ubiquitously expressed member of transient receptor potential (TRP) superfamily (*Nadler et al., 2001*; *Ramsey et al., 2006*; *Venkatachalam and Montell, 2007*), is a cation channel fused with an unique alpha-kinase domain on its C-terminus (*Nadler et al., 2001*). TRPM7 is able to transport divalent cations, particularly $Mg^{2+}$ and $Ca^{2+}$ (*Monteilh-Zoller et al., 2003*). TRPM7 is involved in physiological and pathological processes, including embryonic development, organogenesis, and organism survival (*Jin et al., 2008*; *Ryazanova et al., 2010*). The cellular functions of TRPM7 have been extensively studied in non-neuronal cells (*Abumaria et al., 2018*). The kinase domain of TRPM7 is believed to be important for cell growth and proliferation through either eEF2-kinase activation (*Perraud et al., 2011*) or chromatin remodeling (*Krapivinsky et al., 2014*). On the other hand, it has been suggested that the ion channel region of TRPM7 may be critical for cell survival by regulating $Mg^{2+}$ homeostasis within cells, such as DT40 cell lines and embryonic stem cells (*Ryazanova et al., 2010*; *Schmitz et al., 2003*).

TRPM7 is widely distributed in the nervous system (*Grube et al., 2003*; *Krapivinsky et al., 2006*), and it may regulate animal behaviors, such as learning and memory in mouse and rat (*Liu et al., 2018*) and escape response in zebrafish (*Low et al., 2011*). A TRPM7 mutation with a defective ion channel

property has been identified in human patients with Guamanian amyotrophic lateral sclerosis and Parkinson's disease (*Hermosura et al., 2005*), indicating a link between TRPM7 and neurodegenerative diseases. For the cellular physiology of neurons, TRPM7's kinase domain is reported to be important for normal synaptic density and long-term synaptic plasticity in excitatory neurons from the central nervous system (CNS) (*Liu et al., 2018*). The ion channel region of TRPM7 may regulate acetylcholine release in sympathetic neurons from the peripheral nervous system (PNS) (*Krapivinsky et al., 2006*; *Montell, 2006*), while the role of TRPM7 as an ion channel in CNS neurons has been focused on pathological neuronal death. TRPM7 may mediate neuronal $Ca^{2+}$ overload, resulting in neuronal death under prolonged oxygen/glucose deprivation in neuronal culture (*Aarts et al., 2003*). Meanwhile, knocking down TRPM7 prevents neuronal death induced by either anoxia (*Aarts et al., 2003*) or ischemia (*Sun et al., 2009*). Furthermore, pharmacological blockage of TRPM7 reduces brain damage during neonatal hypoxic injury (*Chen et al., 2015*). TRPM7 as an ion channel is therefore considered as a potential molecular target for drug design to prevent neuronal death (*Bae and Sun, 2013*). Yet, the physiological role of the TRPM7's ion channel region in CNS neurons remains unexplored.

A critical function of TRPM7 in membrane trafficking has been highlighted from non-neuronal cells (*Abiria et al., 2017*). Along this line, TRPM7 has been implicated in presynaptic vesicle recycling in both neuroendocrine cells (*Brauchi et al., 2008*) and PNS neurons (*Krapivinsky et al., 2006*). In PNS sympathetic neurons (*Krapivinsky et al., 2006*), it is reported that TRPM7 may regulate release of positively charged acetylcholine during vesicle fusion through a proposed ion compensation mechanism (*Krapivinsky et al., 2006*; *Montell, 2006*). However, it remains largely unknown what the physiological function of presynaptic TRPM7 is in CNS neurons. By using a combination of biophysical, molecular biology, electrophysiological, and live-cell imaging methods, the present study has identified that $Ca^{2+}$ influx via TRPM7 may be critical for short-term changes of synaptic strength by regulating synaptic vesicle endocytosis in CNS neurons.

## Results

### Endocytic kinetics is slower in TRPM7 KO chromaffin cells

To understand potential roles of TRPM7 in exocytosis and endocytosis, TRPM7 knockout (KO) chromaffin cells were harvested from conditional KO newborn pups containing both *Trpm7[fl/fl]* (*Jin et al., 2008*) and *Th-Cre* (a Cre specific to catecholaminergic cells; *Savitt et al., 2005*) genes. The depletion of TRPM7 expression in medulla of adrenal glands was further confirmed by RT-PCR (*Figure 1—figure supplement 1A*). Single vesicle endocytosis was monitored in neuroendocrine chromaffin cells using cell-attached capacitance recordings with a millisecond time resolution, and representative endocytic events from wild-type (WT) or TRPM7 KO cells were shown in *Figure 1A*. Data from cell-attached capacitance recordings (*Figure 1A*) showed that, while the number of endocytic events (*Figure 1B*), the capacitance size of endocytic vesicles (*Figure 1C*), and the fission-pore conductance (Gp) (*Figure 1D*) were indistinguishable between WT and KO cells, the fission-pore duration (*Figure 1A and E*; *Figure 1—source data 1*) was significantly increased in TRPM7 KO cells, suggesting a critical role of TRPM7 in regulating endocytic kinetics. This finding was further confirmed with data re-analysis, which has a shorter duration cutoff and thus includes additional endocytic events with fission-pore durations in the range of 5–15 ms (*Figure 1—figure supplement 2*, *Figure 1—figure supplement 2—source data 1*), indicating that the increase in the fission-pore duration may be independent of durations of endocytic events in TRPM7 KO cells.

Since localizations of TRPM7 to synaptic vesicles has been implied from previous studies (*Brauchi et al., 2008*; *Krapivinsky et al., 2006*), to examine any potential TRPM7 openings during endocytosis, we performed double (whole-cell/cell-attached) patch recordings, in which the cell-attached patch pipette was utilized to monitor both endocytic events and the ionic current across the patch membrane. With the cell constantly clamped at −60 mV by the whole-cell patch pipette, the patch of membrane captured by the cell-attached pipette was held at different potentials by varying the voltage in the cell-attached pipette. *Figure 1F* showed at +40 mV an ionic current drift ($I_{patch}$) associated with endocytosis in WT cells, and the I-V relationships of the endocytosis-associated ionic current drift in WT cells (*Figure 1G*, *Figure 1—source data 1*) seemed to display the outwardly rectifying characteristics of whole-cell TRPM7 currents as reported in the literature (*Nadler et al., 2001*).

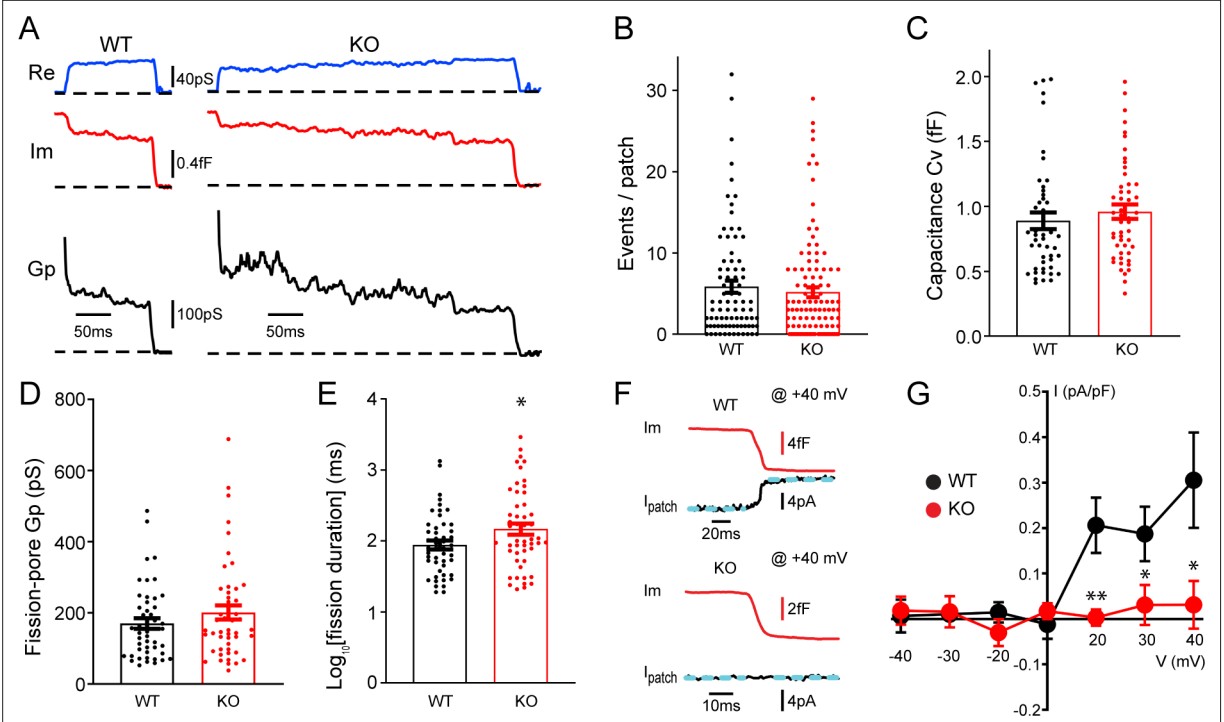

**Figure 1.** Transient receptor potential melastatin 7 (TRPM7) is important for endocytic fission during endocytosis in chromaffin cells. (**A**) Representative endocytic events, as membrane conductance (Re), membrane capacitance (Im), and the fission-pore conductance (Gp), in wild-type (WT) and TRPM7 knockoout (KO) cells. (**B–E**) The number of endocytic events (WT: n = 83 cells; KO: n = 110 cells) (**B**), the capacitance Cv of endocytic vesicles (**C**), and the fission-pore Gp (**D**) were statistically comparable between WT and KO cells; the log-transformed fission-pore duration (**E**) was significantly increased in KO cells (WT: n = 48 events; KO: n = 50 events). (**F**) Representative membrane capacitance (Im) and patch membrane current ($I_{patch}$) traces at +40 mV in WT and KO cells. Dashed lines in cyan represent linear fitting lines before or after capacitance drop induced by endocytosis. (**G**) The current-voltage relationship indicating that the endocytosis-associated current drift is reduced at positive membrane potentials in TRPM7 KO cells. The x-axis represents voltages across the patch membrane. n is 12, 9, 13, 12, 13, 8, and 11 for membrane voltages at –40 , –30 , –20 , 0 , +20 , +30 , and +40 mV in WT cells, and 12, 7, 8, 10, 11, 13, and 11 for membrane voltages at –40 , –30 , –20 , 0 , +20 , +30 , and +40 mV in KO cells, respectively. *$p < 0.05$ and **$p < 0.01$, unpaired two-tailed Student's t-test.

The online version of this article includes the following source data and figure supplement(s) for figure 1:

**Source data 1.** TRPM7 and single vesicle endocytosis in chromaffin cells.

**Figure supplement 1.** Transient receptor potential melastatin 7 (TRPM7) depletion in adrenal medulla and cerebral cortex of the brain and adrenal medulla.

**Figure supplement 2.** Transient receptor potential melastatin 7 (TRPM7) is important for endocytic kinetics in chromaffin cells.

**Figure supplement 2—source data 1.** Increases in the fission-pore duration in KO cells.

**Figure supplement 3.** Endocytosis-associated current drift is independent of the capacitance size of endocytic events.

**Figure supplement 3—source data 1.** Endocytic vesicle size and endocytosis-associated current drift.

**Figure supplement 4.** An ionic current is specifically associated with single vesicle endocytosis.

**Figure supplement 4—source data 1.** Ionic current drifts associated with endocytosis but not exocytosis.

It is important to note that the ionic current drift (*Figure 1F*) is synchronized with the capacitance decrease, due to single vesicle endocytosis (*Yao et al., 2012*), which indicates that ion channels contributing to the current drift may be located on the endocytic vesicle. The upward drift in the membrane current at positive potentials may be due to a loss of an outward ionic current on the endocytic vesicle (*Figure 1F–G*), while the downward drift at negative potentials may be attributed to a loss of an inward ionic current upon the separation of the endocytic vesicle from the plasma membrane (*Figure 1G*). The endocytosis-associated ionic current drift cannot be attributed to a loss of non-specific leak channels on the endocytic vesicle, for the consequence would be predicted to have a linear current-voltage relationship at both negative and positive potentials. On the contrary, the endocytosis-associated ionic current drift we observed displays a non-linear current-voltage relationship (*Figure 1G*). It is of

note that TRPM7-mediated currents have been missed in previous whole-cell recordings in chromaffin cells, which were typically performed, either at its resting membrane potential of around –80 mV (*Smith and Neher, 1997*), or in the presence of 0.5–1 mM $Mg^{2+}$ in the intracellular solution (*Smith and Neher, 1997*; *Wu et al., 2021*) that substantially inhibits TRPM7 currents (*Nadler et al., 2001*).

Quantifications showed that ionic current drifts at positive membrane potentials were substantially reduced in KO cells as compared to WT cells (*Figure 1F–G*), suggesting TRPM7 openings during endocytosis. Meanwhile, the ionic current drift seemed to be associated with single vesicle endocytosis regardless of their capacitance size (*Figure 1—figure supplement 3*, *Figure 1—figure supplement 3—source data 1*). Interestingly, there was no such ionic current drift associated with single vesicle exocytosis in WT cells (*Figure 1—figure supplement 4A*), suggesting openings of TRPM7 channels specifically coincide with endocytosis but not exocytosis. Consistently, the biophysical calculation identified an extra conductance associated with endocytosis rather than exocytosis (*Figure 1—figure supplement 4B, C*, *Figure 1—figure supplement 4—source data 1*).

## Exocytosis remains unaltered in TRPM7 KO chromaffin cells

To determine whether TRPM7 is involved in exocytosis, catecholamine release from single large dense-core vesicles (LDCVs) was analyzed using carbon fiber amperometry. Secretion induced by 90 mM KCl application for 5 s was recorded by a carbon fiber electrode placed in a direct contact with cells (*Figure 2A*). The frequency of amperometrical spikes within the first 15 s of stimulation was indistinguishable between WT and KO cells (*Figure 2B*). Furthermore, no noticeable differences were observed between WT and KO cells for a variety of spike parameters such as peak amplitude (*Figure 2C*), halfwidth (*Figure 2D*), the time constant of falling phase (*Figure 2E*), and quantal size (*Figure 2F*; *Figure 2—source data 1*). Amperometry can also provide detailed millisecond information about the initial release through a transient fusion pore (*Chow et al., 1992*), the foot signal, preceding the subsequent spike of catecholamine release (*Figure 2G*). Our data showed no difference in foot duration between WT and TRPM7 KO cells (*Figure 2H–I*; *Figure 2—source data 1*). Therefore, our amperometrical results indicate that TRPM7 may be dispensable for catecholamine release from LDCVs, which is confirmed by our whole-cell capacitance recordings by detecting exocytosis induced by membrane depolarizations (*Figure 2—figure supplement 1*, *Figure 2—figure supplement 1—source data 1*). Our results are consistent with a previous study showing that TRPM7 knockdown or expression of a dominant negative TRPM7 mutant does not affect LDCVs secretion in PC12 cells (*Brauchi et al., 2008*).

## Synaptic vesicle endocytosis is impaired in neurons from TRPM7 KO animals

To examine potential roles of TRPM7 in synaptic transmission, we cultured TRPM7 KO neurons from conditional KO newborn pups containing both *Trpm7*fl/fl (*Jin et al., 2008*) and *Nestin-Cre* (a brain specific Cre; *Tronche et al., 1999*) genes, and the depletion of TRPM7 in cortical neurons was confirmed by RT-PCR (*Figure 1—figure supplement 1B*). Next, to directly examine synaptic vesicle endocytosis, we performed live-cell imaging of neurons with synaptophysin-pHluorin (sypHy) (*Fernández-Alfonso and Ryan, 2004*; *Granseth et al., 2006*; *Soykan et al., 2017*; *Voglmaier et al., 2006*) expression driven by the human synapsin promoter (*Nathanson et al., 2009*; *Yaguchi et al., 2013*). There was an obvious increase in the time constant of decay in sypHy fluorescent signals in TRPM7 KO neurons at either room temperature (*Figure 3A and B*, *Figure 3—source data 1*) or physiological temperature of 34 °C (*Figure 3—figure supplement 1*, *Figure 3—figure supplement 1—source data 1*), indicative of an endocytic defect of synaptic vesicles. A similar endocytic defect was observed in neurons expressing vGlut1-pHluorin, which revealed a longer time constant for fluorescence decay in KO neurons as well (*Figure 3C–D*, *Figure 3—source data 1*). This endocytic defect cannot be the consequence from any changes in vesicle reacidification, since the reacidification rate of newly formed endocytic vesicles, as demonstrated in *Figure 3—figure supplement 2*, was comparable between WT and KO neurons measured with sypHy (*Figure 3E–F*, *Figure 3—source data 1*). Moreover, the endocytic defects we have observed in TRPM7 KO neurons are unlikely due to any potential alterations in endocytic protein levels, as there was no change in expression levels of several key endocytic proteins we tested, such as dynamin 1, AP2, clathrin, and Syt1, in both chromaffin cells (*Figure 3—figure supplement 3*, *Figure 3—figure supplement 3—source data 1*) and presynaptic

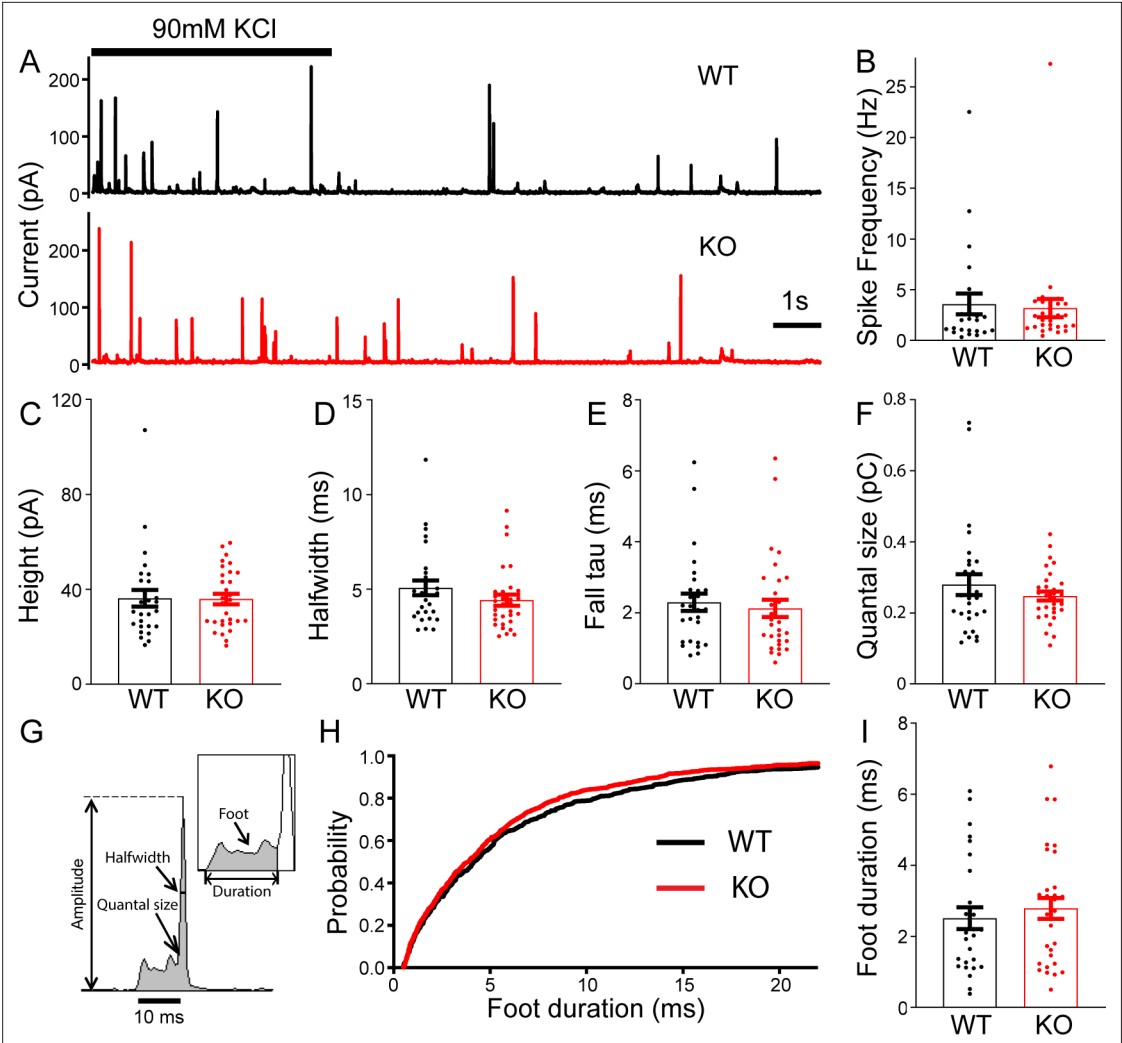

**Figure 2.** No change in large dense-core vesicles (LDCVs) exocytosis in transient receptor potential melastatin 7 (TRPM7) knockout (KO) chromaffin cells detected by carbon fiber amperometry.

(**A**) Amperometrical detection of catecholamine release from wild-type (WT) (upper) and KO (lower) chromaffin cells stimulated with 90 mM KCl for 5 s. Each amperometrical spike represents release from an individual LDCV. (**B–F**) No differences between WT (n = 28) and TRPM7 KO (n = 31) cells in a variety of amperometric parameters, such as spike frequency (**B**), height (**C**), halfwidth (**D**), the exponential time constant of falling phase (**E**), and quantal size (**F**). (**G**) Diagram of the parameters analyzed in amperometric spikes. (**H–I**) Analysis of foot duration. Normalized cumulative distributions of all events in all cells (WT: n = 482; KO: n = 642) (**H**) and average of median values obtained from individual WT (n = 28) and KO (n = 31) cells (**I**). Unpaired two-tailed Student's t-test.

The online version of this article includes the following source data and figure supplement(s) for figure 2:

**Source data 1.** No change in exocytosis in KO chromaffin cells.

**Figure supplement 1.** No change in exocytosis in transient receptor potential melastatin 7 (TRPM7) knockout (KO) chromaffin cells measured by whole-cell capacitance recordings.

**Figure supplement 1—source data 1.** Whole cell capacitance recordings in WT and KO cells.

terminals in neurons (***Figure 3—figure supplement 4***, ***Figure 3—figure supplement 4—source data 1***) from TRPM7 KO animals. Collectively, our data indicates that TRPM7 may be important for synaptic vesicle endocytosis in neurons.

Conversely, paired whole-cell recordings in pyramidal neurons of cortical culture revealed no change in the amplitude of inhibitory postsynaptic currents (IPSCs) evoked by single presynaptic stimulation with 1 ms depolarization to +30 mV in KO neurons (***Figure 3—figure supplement 5A,B***, ***Figure 3—figure supplement 5—source data 1***). There was also no obvious change in the paired pulse ratio (***Figure 3—figure supplement 5C***, ***Figure 3—figure supplement 3—source data 1***),

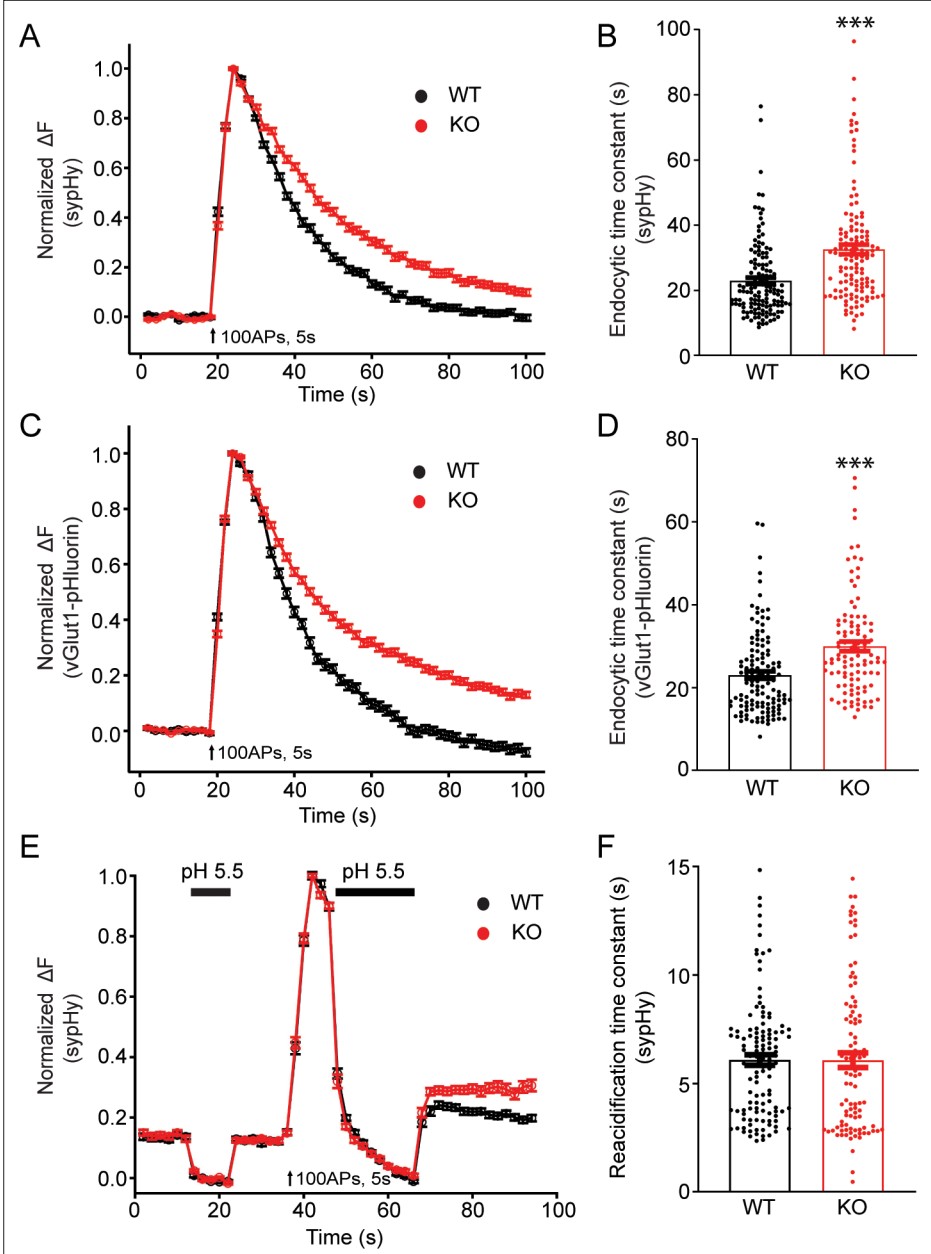

**Figure 3.** Synaptic vesicle endocytosis is impaired in transient receptor potential melastatin 7 (TRPM7) knockout (KO) neurons using pHluorin-based optical imaging assays. (**A**) Normalized fluorescence changes of synaptophysin-pHluorin (sypHy) signals in wild-type (WT) (n = 139) and KO (n = 127) neurons at 2 mM $[Ca^{2+}]_e$. (**B**) Bar graph comparing the endocytic time constant from sypHy experiments in A. (**C**) Normalized fluorescence changes of vGlut1-pHluorin signals in WT (n = 127) and KO (n = 105) neurons at 2 mM $[Ca^{2+}]_e$. (**D**) Bar graph comparing the endocytic time constant from vGlut1-pHluorin experiments in C. (**E**) Normalized sypHy fluorescence traces (WT: n = 121; KO: n = 101) with two periods of perfusion with pH 5.5 MES buffer as indicated in the figure (black bar). Newly endocytosed vesicles 'trapped' during acidic buffer perfusion starting 5 s after 100 action potentials (APs) at 20 Hz. (**F**) Bar graph for the reacidification rate of endocytic vesicles, which was obtained by exponential fittings of fluorescence decay, indicates no change in reacidification in KO neurons. n is the total number of boutons pooled from three animals. ***p < 0.001, unpaired two-tailed Student's t-test.

The online version of this article includes the following source data and figure supplement(s) for figure 3:

**Source data 1.** Defects in synaptic vesicle endocytosis in KO neurons.

**Figure supplement 1.** Endocytic defects of synaptic vesicles in transient receptor potential melastatin 7 (TRPM7) knockout (KO) neurons at physiological temperature of 34 °C.

*Figure 3 continued on next page*

*Figure 3 continued*

**Figure supplement 1—source data 1.** No change in synaptic vesicle endocytosis in KO neurons at physiological temperature.

**Figure supplement 2.** Measurements of reacidification rate of newly endocytosed vesicles from single synaptic terminals.

**Figure supplement 3.** No difference in expression levels of endocytic proteins between wild-type (WT) and transient receptor potential melastatin 7 (TRPM7) knockout (KO) chromaffin cells.

**Figure supplement 3—source data 1.** No change in endocytic proteins expressions in TRPM7 KO chromaffin cells.

**Figure supplement 4.** No change in expression levels of endocytic proteins in presynaptic terminals of transient receptor potential melastatin 7 (TRPM7) knockout (KO) neurons.

**Figure supplement 4—source data 1.** No change in endocytic proteins expressions in presynaptic terminals of KO neurons.

**Figure supplement 5.** Transient receptor potential melastatin 7 (TRPM7) may not be involved in synaptic vesicle exocytosis in inhibitory neurons.

**Figure supplement 5—source data 1.** No change in evoked IPSCs in TRPM7 neurons.

**Figure supplement 6.** No difference in release probability of RRP vesicles between wild-type (WT) and transient receptor potential melastatin 7 (TRPM7) knockout (KO) neurons as measured from vGlut1-pHluorin experiments.

**Figure supplement 6—source data 1.** No difference in release probability between WT and KO neurons.

---

indicating no alteration in release probability in KO neurons. Consistently, vesicular release probability measured in vGlut1-pHluorin experiments was not altered in TRPM7 KO neurons (*Figure 3—figure supplement 6*, *Figure 3—figure supplement 6—source data 1* ). These results suggest that TRPM7 may not be critical for exocytosis of synaptic vesicles in inhibitory synapses with electrically neutral GABA or excitatory synapses with negatively charged glutamate as the predominant neurotransmitters, although it is aware that a previous study implies that TRPM7 may have a post-fusion role by supplying counterions during release of positively charged acetylcholine in PNS sympathetic neurons (*Krapivinsky et al., 2006*; *Montell, 2006*).

## WT TRPM7 (TRPM7^WT), but not a non-conducting TRPM7 mutant (TRPM7^LCF), rescues endocytic defects in TRPM7 KO chromaffin cells and neurons

To further explore whether the ion channel region of TRPM7 is relevant to TRPM7-dependent synaptic vesicle endocytosis, we generated a non-conducting TRPM7 mutant with a loss of ion channel functions (TRPM7^LCF), by mutating highly conserved amino acids located in the sixth putative transmembrane domain (amino acids 1090–1092, NLL mutated to FAP) (*Krapivinsky et al., 2006*). As reported previously (*Krapivinsky et al., 2006*), when compared to TRPM7^WT, there was an almost complete suppression of TRPM7^LCF currents in HEK 293 cells with lentiviral infection (*Figure 4A–B*, *Figure 4—source data 1* and *Figure 4—figure supplement 1*). Using cell-attached recordings, we examined kinetics of single endocytic events in KO chromaffin cells expressing either TRPM7^WT or TRPM7^LCF (*Figure 4C*), with similar TRPM7^WT or TRPM7^LCF expression levels as confirmed by immunostaining (*Figure 4—figure supplement 2*, *Figure 4—figure supplement 2—source data 1*). While the number of endocytic events (*Figure 4D*), the capacitance size of endocytic vesicles (*Figure 4E*), and the fission-pore Gp (*Figure 4F*) were comparable between these two groups, the fission-pore duration was reduced in KO cells expressing TRPM7^WT as compared to TRPM7^LCF (*Figure 4G*; *Figure 4—source data 1*). Consistently, the time constant of sypHy signal decay was shorter in KO neurons expressing TRPM7^WT than TRPM7^LCF from live-cell imaging experiments (*Figure 4H–I*; *Figure 4—source data 1*). These findings thus conclude that TRPM7's ion channel region may be important for synaptic vesicle endocytosis in neurons. Alternatively, since TRPM7's kinase activity could be potentially altered in the TRPM7^LCF mutant, it could be possible that the endocytic defect we observed in KO neurons might reflect roles of TRPM7's C-terminal kinase in synaptic vesicle endocytosis. However, this possibility is unlikely, since annexin A1 (*Dorovkov et al., 2011*), the so far identified substrate of TRPM7 kinase important for endocytosis (*Futter and White, 2007*), is largely expressed in ependymal and glial cells rather than neurons in the brain (*Solito et al., 2008*).

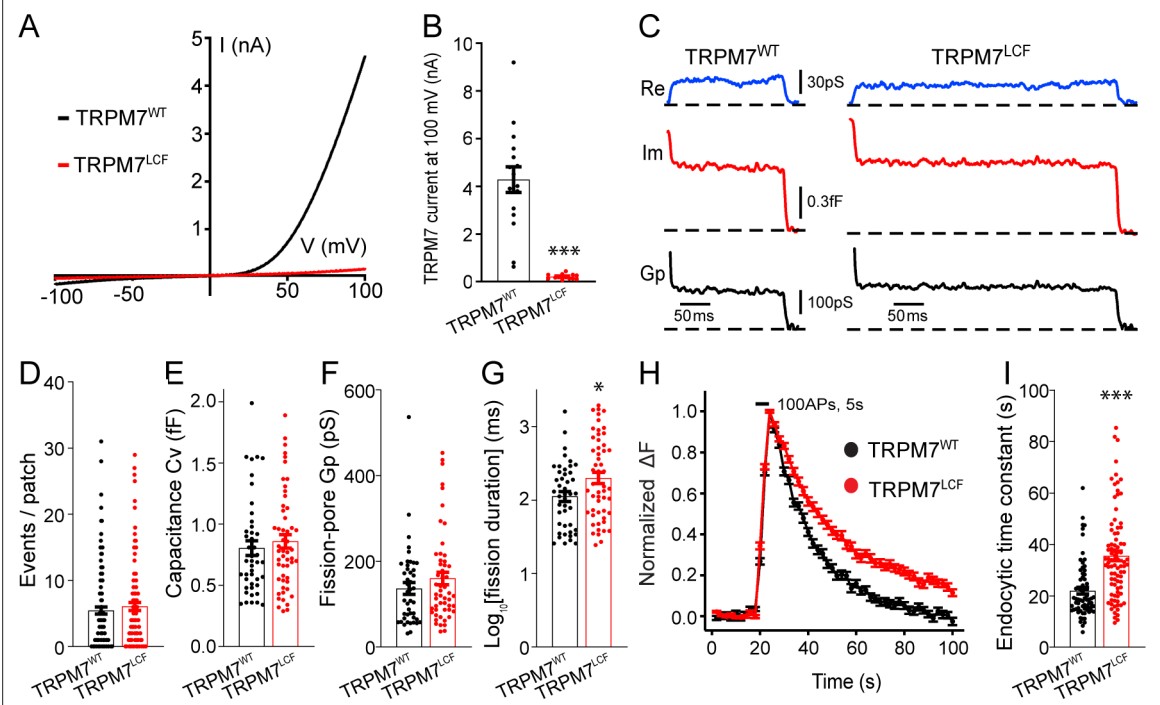

**Figure 4.** Transient receptor potential melastatin 7 (TRPM7) as an ion channel is critical for endocytosis in both chromaffin cells and neurons. (**A**) Averaged whole-cell TRPM7 currents, recorded from HEK 293 cells transduced with lentivirus encoding TRPM7^WT or TRPM7^LCF, in response to voltage ramps from –100 to +100 mV. (**B**) Quantifications showing TRPM7^LCF currents (n = 16 cells) at +100 mV were almost completely blocked as compared to TRPM7^WT currents (n = 13 cells). (**C**) Representative endocytic events, as membrane conductance (Re), capacitance (Im), and the fission-pore Gp traces in KO cells expressing either TRPM7^WT (left) or TRPM7^LCF (right). (**D–G**) The number of endocytic events (TRPM7^WT: n = 141 cells; TRPM7^LCF: n = 128 cells) (**D**), the capacitance Cv of endocytic vesicles (**E**), and the fission-pore Gp (**F**) were indistinguishable between these two groups; the log-transformed fission-pore duration (**G**) was significantly increased in TRPM7^LCF-expressing KO cells (TRPM7^WT: n = 44 events; TRPM7^LCF: n = 53 events). (**H–I**) Normalized fluorescence changes of synaptophysin-pHluorin (sypHy) signals (**H**) and bar graph (**I**), comparing the endocytic time constant from sypHy experiments in KO neurons expressing TRPM7^WT (n = 69) or TRPM7^LCF (n = 87) neurons, indicate that endocytic kinetics is slower in TRPM7^LCF-expressing KO neurons. n is the total number of boutons pooled from six animals. *p < 0.05 and ***p < 0.001, unpaired two-tailed Student's t-test.

The online version of this article includes the following source data and figure supplement(s) for figure 4:

**Source data 1.** Lentivirus-mediated TRPM7 versions in HEK 293 cells.

**Figure supplement 1.** Confocal images for HEK 293 cells infected with lentivirus encoding with either empty vector, TRPM7^WT or TRPM7^LCF.

**Figure supplement 2.** Similar expression levels of TRPM7^WT and TRPM7^LCF in knockout (KO) chromaffin cells.

**Figure supplement 2—source data 1.** Lentivirus-mediated TRPM7 versions in chromaffin cells.

## The presynaptic Ca²⁺ increase upon a train of stimulations is reduced in TRPM7 KO neurons

Our sypHy experiments showed that there was no difference in the endocytic time constant of fluorescent signal decay between WT and KO neurons perfused with 0 mM $[Ca^{2+}]_e$ right after the cessation of stimulations (***Figure 5A–B***, ***Figure 5—source data 1***), suggesting a role of TRPM7 in $Ca^{2+}$-dependent rather than $Ca^{2+}$-independent synaptic vesicle endocytosis. Therefore, it is speculated that $Ca^{2+}$ influx via TRPM7 may be critical for synaptic vesicle endocytosis. To test this idea, we compared presynaptic $Ca^{2+}$ signaling between WT and TRPM7 KO neurons by using synaptophysin-GCaMP6f (SyGCaMP6f) fusion protein as a presynaptic $Ca^{2+}$ reporter (***de Juan-Sanz et al., 2017***). While neurons responded to electrical stimuli with a robust increase in SyGCaMP6f fluorescence (***Figure 5C–D***), the increase in the SyGCaMP6f signal was substantially reduced in response to a stimulation of 100 action potentials (APs) at 20 Hz in TRPM7 KO neurons (***Figure 5C–D***, ***Figure 5—source data 1***). The reduction in $Ca^{2+}$ rise in response to stimulation train is unlikely due to any changes in $Ca^{2+}$ influx via voltage-gated $Ca^{2+}$ channels (VGCCs), since the peak of SyGCaMP6f signals induced by single AP (***Brockhaus et al., 2019***; ***Kim and Ryan, 2013***) was indistinguishable between WT and KO neurons (***Figure 5E–F***,

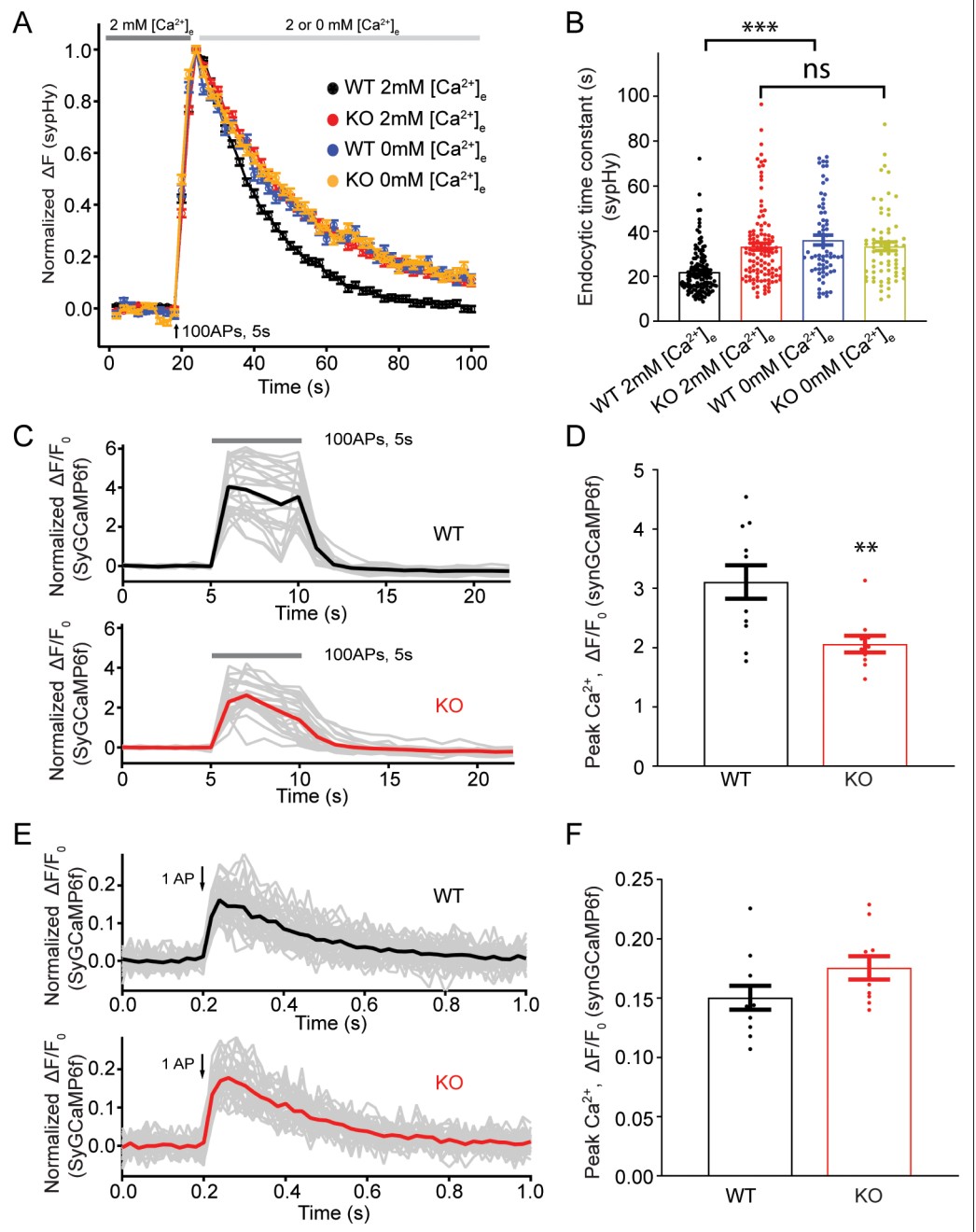

**Figure 5.** Transient receptor potential melastatin 7 (TRPM7) is important for presynaptic $Ca^{2+}$ increase upon a train of stimulations. (**A**) Normalized fluorescence changes of synaptophysin-pHluorin (sypHy) signal in wild-type (WT) (2 mM: n = 131 buttons; 0 mM: n = 68 buttons) and TRPM7 knockout (KO) (2 mM: n = 116 buttons; 0 mM: n = 67 buttons) neurons at 2 and 0 mM $[Ca^{2+}]_e$ after the cessation of stimuli. (**B**) Bar graph comparing the endocytic time constant from sypHy experiments in A. (**C**) Sample traces showing normalized changes in SyGCaMP6f fluorescence stimulated with 100 action potentials (APs) at 20 Hz from an individual coverslip of WT (top) (25 regions of interest [ROIs]) or TRPM7 KO (bottom) (25 ROIs) neurons at 2 mM $[Ca^{2+}]_e$. Gray lines are changes for individual ROIs with the black or red line as the averaged response. (**D**) Bar graph comparing peak values of $Ca^{2+}$ transients from SyGCaMP6f experiments in (C) displays a significant decrease of $Ca^{2+}$ signals in TRPM7 KO neurons (WT: n = 11 coverslips; KO: n = 10 coverslips). (**E**) Sample traces showing normalized changes in SyGCaMP6f fluorescence induced by single AP from an individual coverslip of WT (top) (36 ROIs) or TRPM7 KO (36 ROIs) neurons at 2 mM $[Ca^{2+}]_e$. Gray lines are changes of individual ROIs with black or red line as the averaged response. (**F**) Bar graph comparing peak values of $Ca^{2+}$ transients as shown in (E) revealed no change in $Ca^{2+}$ signals induced by a single

*Figure 5 continued on next page*

*Figure 5 continued*

AP (WT: n = 11 coverslips; KO: n = 10 coverslips). **p < 0.01, unpaired two-tailed student's t-test; ***p < 0.001, Newman of one-way ANOVA in (B).

The online version of this article includes the following source data and figure supplement(s) for figure 5:

**Source data 1.** Decreased presynaptic Ca2+ signals during stimulation train.

**Figure supplement 1.** No difference in the resting $Ca^{2+}$ concentration between wild-type (WT) and transient receptor potential melastatin 7 (TRPM7) knockout (KO) chromaffin cells.

**Figure supplement 1—source data 1.** No difference in the resting Ca2+ concentration between WT and KO chromaffin cells.

**Figure supplement 2.** No change in the basal $Ca^{2+}$ concentration in presynaptic terminals of transient receptor potential melastatin 7 (TRPM7) knockout (KO) neurons.

**Figure supplement 2—source data 1.** No difference in the resting Ca2+ concentration of synaptic terminals between WT and KO neurons.

**Figure supplement 3.** The decay of $Ca^{2+}$ signals after the cessation of the train stimulation is comparable between wild-type (WT) and transient receptor potential melastatin 7 (TRPM7) knockout (KO).

**Figure supplement 3—source data 1.** Similar decay time constant of SyGCaMP6f siganls after stimulation train in WT and KO neurons.

*Figure 5—source data 1*). Meanwhile, the resting $Ca^{2+}$ concentration remained unaltered in both chromaffin cells (*Figure 5—figure supplement 1*, *Figure 5—figure supplement 1—source data 1*) and presynaptic terminals in neurons from TRPM7 KO animals (*Figure 5—figure supplement 2*, *Figure 5—figure supplement 2—source data 1*), and the time constant of $Ca^{2+}$ signal decay right after the cessation of stimulations was similar between WT and TRPM7 KO neurons (*Figure 5—figure supplement 3*, *Figure 5—figure supplement 3—source data 1*).

## Short-term depression of synaptic transmission is enhanced in TRPM7 KO neurons

Our data presented so far indicates that $Ca^{2+}$ influx via TRPM7 may be critical for synaptic vesicle endocytosis. Since synaptic vesicle endocytosis is an essential presynaptic factor for short-term synaptic plasticity (*Regehr, 2012*; *Zucker and Regehr, 2002*), we next examined whether short-term synaptic depression is altered in TRPM7 KO neurons.

For inhibitory synaptic transmission, evoked IPSCs in pyramidal neurons were recorded by stimulating presynaptic inhibitory neurons with 250 stimuli at 10 Hz through a presynaptic whole-cell pipette (*Figure 6A*). With the amplitude of evoked IPSCs normalized to the first response, KO neurons displayed a more profound short-term synaptic depression (*Figure 6A–B*) and a 56 % reduction of the steady-state IPSCs (insert in *Figure 6B*, *Figure 6—source data 1*), suggesting an important role of TRPM7 in the short-term synaptic depression of inhibitory synaptic transmission.

For excitatory synaptic transmission, high-speed optical imaging of evoked glutamate release using iGluSnFR (*Marvin et al., 2018*) was utilized to directly monitor the presynaptic short-term plasticity (*Figure 6C*). Neurons on coverslips with iGluSnFR expression were challenged with 50 stimuli at 10 Hz to induce short-term depression of evoked glutamate release, fluorescent healthy dendrite arbors were focused and imaged at 50 Hz (*Figure 6C–D*). Similar to that observed in the synaptic plasticity measured with evoked IPSCs (*Figure 6A–B*), normalized changes of the iGluSnFR fluorescence during the train also revealed an enhanced short-term depression in TRPM7 KO neurons (*Figure 6D–E*). The normalized steady-state iGluSnFR signals at the end of the train was reduced by 36 % in TRPM7 KO neurons (p < 0.001) (insert in *Figure 6E*, *Figure 6—source data 1*), implying a significance of TRPM7 in short-term depression of excitatory synaptic transmission.

Taken together, the enhanced presynaptic short-term depression in both inhibitory and excitatory neurons from TRPM7 KO animals (inserts in *Figure 6B and E*) corroborated each other. Given the well-established significance of synaptic vesicle endocytosis in short-term depression (*Henkel and Betz, 1995*; *Hermosura et al., 2005*), our data may suggest that TRPM7 is critical for presynaptic short-term synaptic depression via its regulation in synaptic vesicle endocytosis.

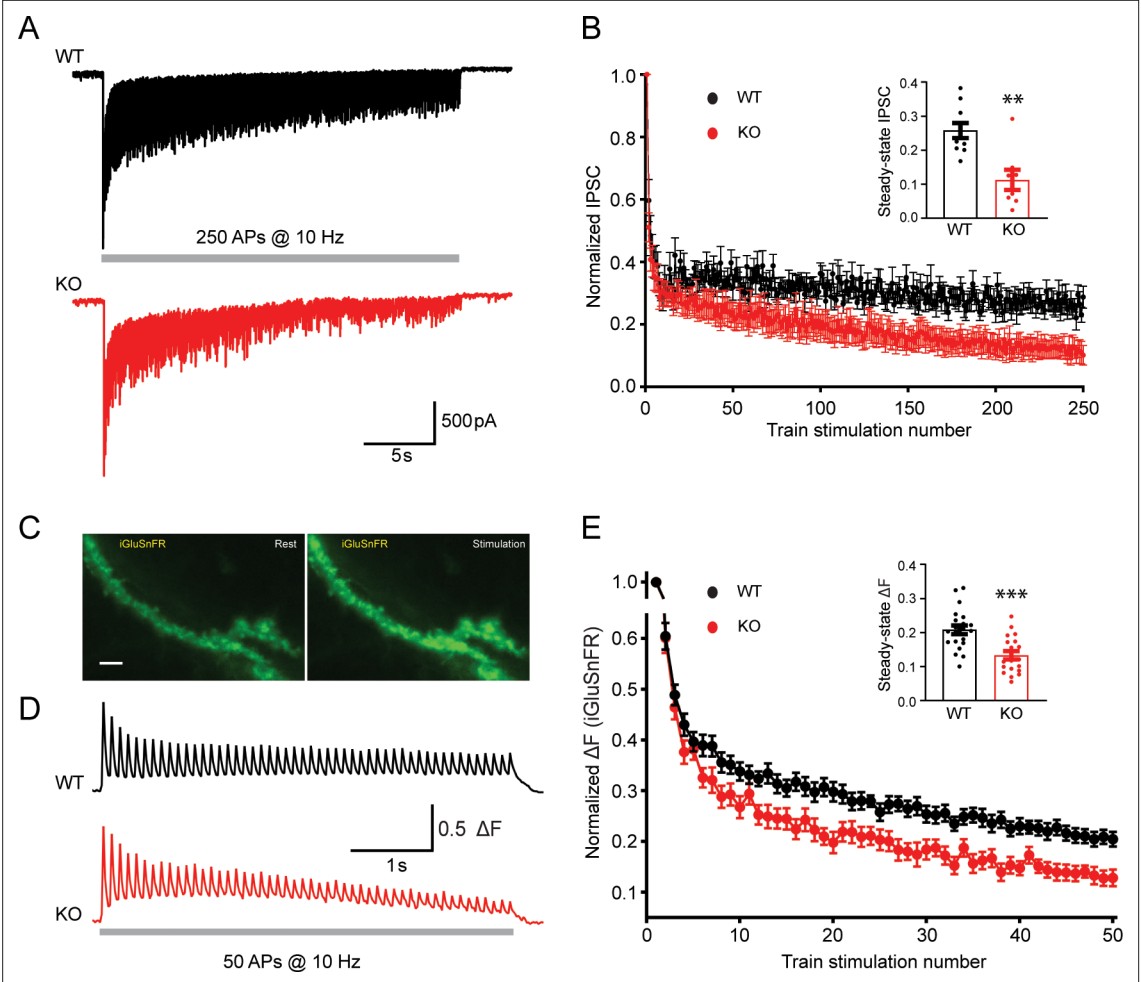

**Figure 6.** Transient receptor potential melastatin 7 (TRPM7) is critical for presynaptic short-term depression in both inhibitory and excitatory synaptic transmissions. (**A–B**) Representative traces of inhibitory postsynaptic currents (IPSCs) evoked by 250 stimulations at 10 Hz in wild-type (WT) (top in A) and TRPM7 knockout (KO) (bottom in A) neurons (**A**) and graph of averaged normalized IPSCs changes (**B**). Bar graph of the steady-state IPSCs calculated as the average of last 10 evoked responses in WT and TRPM7 KO neurons are shown in insert in B. Number of recordings is 10 for WT and 8 for KO. (**C**) Representative images showing fluorescence levels of the same dendritic arbor expression iGluSnFR at rest (left) or upon stimulations (right). Scale bar, 5 µm. (**D–E**) Representative traces, from WT (top in D) and TRPM7 KO (bottom in D) neurons (**D**), and normalized graphs (**E**) of changes in iGluSnFR fluorescence (ΔF/ΔFmax) evoked by 50 stimulations at 10 Hz. Bar graph of the steady state of iGluSnFR ΔF/ΔFmax calculated as the average of last five evoked responses in WT and TRPM7 KO neurons are shown in insert in (E). Number of coverslips is 21 for WT and 20 for KO. **p < 0.01 and ***p < 0.001, unpaired two-tailed Student's t-test.

The online version of this article includes the following source data for figure 6:

**Source data 1.** Enhanced short-term depression in KO neurons.

## Discussion
### TRPM7 may serve as a Ca²⁺ influx pathway for synaptic vesicle endocytosis

With VGCCs as the well-established $Ca^{2+}$ influx channels for synaptic vesicle exocytosis, the identity of $Ca^{2+}$ influx channels for endocytosis remains uncertain. Since VGCCs have been shown to be important for endocytosis (*Midorikawa et al., 2014*; *Perissinotti et al., 2008*; *Xue et al., 2012*), it is implied that $Ca^{2+}$ for endocytosis at the periactive zone may be derived from diffusion of VGCCs-mediated $Ca^{2+}$ influxes at active zones. On the other hand, exocytosis can be triggered, without activations of VGCCs, by black widow spider venom (*Ceccarelli and Hurlbut, 1980*; *Henkel and Betz, 1995*), caffeine (*Zefirov et al., 2006*), or veratridine (*Kuromi and Kidokoro, 2002*). Interestingly, extracellular $Ca^{2+}$ is still necessary for synaptic vesicle endocytosis in these paradigms, suggesting

distinct $Ca^{2+}$ channels independent of VGCCs for endocytosis. Indeed, a $La^{3+}$-sensitive $Ca^{2+}$ channel is indicated to be required for synaptic vesicle endocytosis in *Drosophila* (***Kuromi et al., 2004***).

While it has been proposed that a $Ca^{2+}$-permeable Flower channel may be necessary for synaptic vesicle endocytosis at the neuromuscular junction in *Drosophila* (***Yao et al., 2009***), the function of Flower as a $Ca^{2+}$ influx channel in synaptic vesicle endocytosis of mammalian systems remains questionable, since the role of this $Ca^{2+}$-permeable protein in synaptic vesicle endocytosis was not confirmed at the calyx of Held (***Xue et al., 2012***). Additionally, a recent study showed that the protein itself, but not the ion channel property of Flower, is important for synaptic vesicle endocytosis elicited by moderate stimulations in rat hippocampal neurons (***Yao et al., 2017***). Furthermore, the time course of $Ca^{2+}$ influx via Flower protein, in the order of 10–60 min (***Yao et al., 2009***), may be too slow for synaptic vesicle endocytosis.

Our results showed that $Ca^{2+}$ increase during stimulation train is reduced in TRPM7 KO neurons (***Figure 5C–D***), indicating an importance of TRPM7-mediated $Ca^{2+}$ influx in maintaining $Ca^{2+}$ levels during stimulations. Alternatively, this reduction could be due to slower VGCCs depression and/or slower $Ca^{2+}$ extrusion mechanisms during stimulation train. On the other hand, sypHy experiments indicate that post-stimulus $Ca^{2+}$ influx may be critical for the TRPM7-mediated synaptic vesicle endocytosis (***Figure 5A–B***). Interestingly, $Ca^{2+}$ signals returned to the resting level quickly, with time constant ~0.6 s, right after the cessation of stimulations, which is similar as described in the literature (***Akerboom et al., 2012***; ***Brockhaus et al., 2019***; ***Sgobio et al., 2014***; ***Singh et al., 2018***), and there was no alteration in the resting $Ca^{2+}$ level in presynaptic terminals in TRPM7 KO neurons (***Figure 5— figure supplement 2***). Taken together, it is reasonable to consider that TRPM7 may regulate local $Ca^{2+}$ levels in specific presynaptic regions close to the periactive zone, sites for synaptic vesicle endocytosis (***Saheki and De Camilli, 2012***), although the global $Ca^{2+}$ level as we have detected within presynaptic terminals is quickly recovered to the resting level after stimulations. However, measurements of local $Ca^{2+}$ signals of a specific region within presynaptic terminals are challenging for techniques currently utilized in the field.

Collectively, our data suggest that TRPM7 channels may serve a $Ca^{2+}$ influx pathway for synaptic vesicle endocytosis in neurons. Along this line, it is of note that $Ca^{2+}$ influx via TRPM7 may be essential for endocytosis of toll-like receptor 4 in the non-neuronal microphage (***Schappe et al., 2018***). A recent study indicates that TRPC5, another TRP superfamily member, may serve as a $Ca^{2+}$ influx pathway for synaptic vesicle endocytosis in hippocampal neurons (***Schwarz et al., 2019***). Since TRP channels are widely expressed in the brain (***Kunert-Keil et al., 2006***), it is speculated that members of the TRP superfamily may serve as general $Ca^{2+}$ channels for synaptic vesicle endocytosis in different regions of the nervous system.

## $Ca^{2+}$ influx via TRPM7 may regulate the kinetics rather than the number of single endocytic events

Vesicle fission is a crucial step during endocytosis (***Kaksonen and Roux, 2018***). The importance of vesicle fission in endocytosis can be reflected by significant roles of the GTPase dynamin, a key molecule essential for vesicle fission (***Antonny et al., 2016***), in synaptic vesicle endocytosis (***Ferguson et al., 2007***; ***Newton et al., 2006***). Our previous work using cell-attached capacitance recordings has shown that inhibition of dynamin GTPase slows down endocytic kinetics of single vesicles in chromaffin cells at the millisecond time scale. Therefore, our data shows an increase in the endocytic kinetics of single vesicles in TRPM7 KO chromaffin cells, indicating a potential role of TRPM7 in vesicle fission during endocytosis. The physiological significance of TRPM7 in vesicle fission during endocytosis is reiterated by a slower rate of synaptic vesicle endocytosis in TRPM7 KO neurons.

The rate of synaptic vesicle endocytosis is determined by the number of vesicles involved and the kinetics of single endocytic events. Our results, showing slower kinetics but no change in the number of single endocytic events in TRPM7 KO chromaffin cells, indicate that $Ca^{2+}$ influx via TRPM7 may regulate the rate of synaptic vesicle endocytosis by regulating the kinetics of single endocytic events but not the number of vesicles involved. This result is in line with our previous study, which demonstrates an importance of extracellular $Ca^{2+}$ for the kinetics but not the number of single vesicle endocytosis (***Yao et al., 2012***). Consistently, extracellular $Ca^{2+}$ may modulate endocytic kinetics of single synaptic vesicles in neurons, although no analysis was performed on whether $Ca^{2+}$ affects the number of synaptic vesicles involved (***Leitz and Kavalali, 2011***).

## TRPM7 may be a vesicular protein in presynaptic terminals

By simultaneously monitoring ionic current and membrane capacitance in neuroendocrine chromaffin cells, we have identified an endocytosis-associated ionic current drift, which is substantially reduced in TRPM7 KO cells (*Figure 1F–G*). Since the endocytosis-associated ionic current drifts rather than displaying a pattern of single channel openings (*Figure 1F*), we reason that the endocytosis-associated ionic current drift may be attributed to a loss of multiple opening ion channels with small conductance on the endocytic vesicle upon its separation from the plasma membrane. Consistently, our data shows that the endocytosis-associated current drift is synchronized with capacitance drop, which reflects a separation of the endocytic vesicle from the plasma membrane. In fact, TRPM7 is a channel that can be activated by $PI(4,5)P_2$ and inactivated upon $PI(4,5)P_2$ hydrolysis (*Runnels et al., 2002*). Interestingly, $PI(4,5)P_2$ is concentrated on the plasma membrane but absent from synaptic vesicles, due to $PI(4,5)P_2$ hydrolysis by phosphatases such as synaptojanin 1 during uncoating after vesicle pinch-off, a late stage of endocytosis (*Cremona and De Camilli, 2001*). Therefore, the specific openings of TRPM7 during endocytosis may be attributed to the tight coupling between $PI(4,5)P_2$ metabolism and synaptic vesicle recycling (*Cremona and De Camilli, 2001*).

Consistent with our data, previous studies have indicated TRPM7 as a vesicular protein in non-neuronal cells (*Abiria et al., 2017*), neuroendocrine cells (*Brauchi et al., 2008*), and neurons (*Krapivinsky et al., 2006*). Immunostaining in PC12 cells, a cell line of neuroendocrine chromaffin cells, has found that TRPM7 is localized to small synaptic-like vesicles (*Brauchi et al., 2008*). A biochemical analysis of purified rat brain synaptosomes and synaptic vesicles supports this finding by revealing that TRPM7 is concentrated in those regions but absent from postsynaptic densities (*Krapivinsky et al., 2006*). Further, the same study suggests that TRPM7 may form a molecular complex with vesicular proteins such as SYN1 and synaptotagmin 1 (*Krapivinsky et al., 2006*). On the other hand, a systematic survey of synaptic vesicle proteins did not detect TRPM7 as a synaptic vesicle protein (*Takamori et al., 2006*). However, this survey has missed some well-known synaptic vesicle proteins, such as SVOP, the chloride channels ClC3 and ClC7, and ever the putative vesicular neurotransmitter transporters (VMATs and VACHT). Indeed, as claimed by the authors, their proteomics results in an overall coverage of ~80 % of all known vesicle membrane proteins (*Takamori et al., 2006*). Therefore, it is possible that the systematic survey of synaptic vesicle proteins may have failed to detect TRPM7 on synaptic vesicles.

## Potential roles of TRPM7 in short-term synaptic depression

Our results indicate an important role of TRPM7 in short-term synaptic depression in both excitatory and inhibitory neurons (*Figure 6*). It is notable that the onset of short-term depression may be different from electrophysiological recordings and iGluSnFR-based live-cell imaging experiments. As compared to WT neurons, the depression in TRPM7 KO neurons seemed to be apparent after ~15 stimuli in electrophysiological recordings (*Figure 6B*), while the depression started to show up earlier after ~5 stimuli from iGluSnFR experiments (*Figure 6E*). This discrepancy could be due to different sensitivities between these two approaches. Alternatively, it may indicate a slight difference in synaptic vesicle recycling between excitatory neurons from our imaging experiments and inhibitory neurons from electrophysiological experiments. Since it is postulated that clearance of endocytic components from release sites is an important factor for short-term depression (*Hua et al., 2013*; *Neher, 2010*), the enhanced short-term depression we have observed in TRPM7 KO neurons may indicate a potential role of TRPM7 in release site clearance, in addition to synaptic vesicle endocytosis as we have proposed. It is also reported that the degree of short-term depression can be modulated by exocytic parameters, such as release probability and recruitment of release-ready vesicles (*Neher and Sakaba, 2008*). While our data showed no difference in the release probability between WT and TRPM7 KO neurons (*Figure 3—figure supplement 6*), a reduction in presynaptic $Ca^{2+}$ signals during sustained activity in TRPM7 KO neurons (*Figure 5C–D*) could slow down recruitment of release-ready vesicles, by reducing activations of vesicle priming and $Ca^{2+}$ sensors in fusion (*Chapman, 2008*; *Pang and Südhof, 2010*), which could also lead to an enhanced short-term depression.

## The physiological role of TRPM7 in CNS neurons

As an ion channel fused with a C-terminal alpha-kinase (*Nadler et al., 2001*), TRPM7's ion channel region may be critical for synaptic vesicle endocytosis (the present study) while its kinase domain may

be important for normal synaptic density (*Liu et al., 2018*) in CNS neurons under physiological conditions. It is reported that TRPM7 may be involved in learning and memory in mouse and rat, as knockdown or conditional KO of TRPM7 impairs spatial learning (*Liu et al., 2018*). The authors indicate that the role of TRPM7 in learning and memory could reflect its importance in regulating synaptic density and long-term synaptic plasticity in excitatory neurons (*Liu et al., 2018*). However, this interpretation is conflicting with a previous study showing that TRPM7 knockdown does not change evoked synaptic responses, indicative of no change in synaptic density and LTP (long-term potentiation) (*Sun et al., 2009*). Interestingly, recent studies suggest that TRPC5, which regulates synaptic vesicle endocytosis through a mechanism (*Schwarz et al., 2019*) similar to TRPM7 as we have observed in the present study, is critical for both short-term synaptic plasticity in excitatory neurons (*Schwarz et al., 2019*) and learning and memory of animals (*Bröker-Lai et al., 2017*). Therefore, the role of TRPM7 in short-term synaptic plasticity in excitatory neurons identified here may presumably account for its functions in learning and memory of animals as reported previously (*Liu et al., 2018*). Notably, physiological roles of TRPM7 in inhibitory neurons remain largely unknown, although TRPM7 is indeed expressed in both excitatory and inhibitory neurons from single cell transcriptome (*Huntley et al., 2020*). Our study identifies that TRPM7 is also important for short-term synaptic plasticity in inhibitory neurons. Given the increasing significance of inhibitory neuronal circuits in learning and memory (*Barron et al., 2017*), the reported roles of TRPM7 in learning and memory (*Liu et al., 2018*) could, alternatively, be attributed to its modulations of inhibitory synaptic transmission.

## Implications

In summary, our study indicates that $Ca^{2+}$ influx via TRPM7 is critical for synaptic vesicle endocytosis and consequently short-term synaptic depression, thus implying that TRPM7 may be critical for filtering and gain control of synaptic transmission, as short-term synaptic plasticity is an important factor of synaptic computation (*Abbott and Regehr, 2004*).

During electrophysiological recordings, TRPM7 can be fully inhibited by intracellular $Mg^{2+}$ (*Nadler et al., 2001*), which justifies our choice of whole-cell pipette solutions with no $Mg^{2+}$ for electrophysiological recordings in both chromaffin cells and HEK 293 cells. However, roles of influx of ions, such as $Ca^{2+}$, via TRPM7 in neurons have been documented under both physiological (*Krapivinsky et al., 2006*) and pathological (*Aarts et al., 2003*; *Sun et al., 2009*) conditions, indicating that additional cytoplasmic signaling molecules, which remains unknown so far, may be able to facilitate TRPM7 to bypass tonic inhibitions by intracellular $Mg^{2+}$ within intact neurons. In addition, TRPM7 can be blocked by extracellular $Mg^{2+}$ (*Nadler et al., 2001*), and this $Mg^{2+}$-dependent TRPM7 blockage may be relieved upon membrane depolarizations (*Ramsey et al., 2006*), a mechanism similar to postsynaptic NMDA receptors (*Hunt and Castillo, 2012*). Therefore, it is likely that membrane depolarizations will enhance $Ca^{2+}$ influx via TRPM7 and thus accelerate synaptic vesicle endocytosis. Interestingly, many neurons, such as cortical neurons, frequently generate bursts of action potentials in vivo (*Baranyi et al., 1993*; *Gray and McCormick, 1996*), and these bursts can occur at a very high frequency, up to 300 Hz, in awake animals as reported (*Gray and Viana Di Prisco, 1997*). Since synaptic vesicle endocytosis becomes a limiting factor for synaptic transmission in neurons during high-frequency action potential firings (*Kawasaki et al., 2000*), it is likely that neurons may require an up-regulation of synaptic vesicle endocytosis to maintain synaptic transmission. Therefore, it is speculated that roles of TRPM7 in synaptic vesicle endocytosis will be enhanced during high-frequency firings, which may couple synaptic vesicle endocytosis to neuronal activities.

Recent studies have revealed a tight link between neurodegenerative diseases, particularly Parkinson's disease, and proteins that participate in synaptic vesicle endocytosis, such as auxilin (*Song et al., 2017*), synaptojanin 1 (*Cao et al., 2017*), and endophilin (*Trempe et al., 2009*). Our evidence for a role of TRPM7 in synaptic vesicle endocytosis suggests that the consequence on synaptic vesicle endocytosis attributed to TRPM7 dysfunctions could be a potential molecular mechanism for TRPM7-related neurodegenerative diseases, such as Guamanian amyotrophic lateral sclerosis and Parkinson's disease (*Hermosura et al., 2005*).

# Materials and methods

**Key resources table**

| Reagent type (species) or resource | Designation | Source or reference | Identifiers | Additional information |
|---|---|---|---|---|
| Strain, strain background (*Mus musculus*) | TRPM7$^{fl/fl}$ | Jackson Laboratory Stock No: 018784 | RRID:IMSR_JAX:018784 | |
| Strain, strain background (*Mus musculus*) | Nestin-Cre | Jackson Laboratory Stock No: 003771 | RRID:IMSR_JAX:003771 | |
| Strain, strain background (*Mus musculus*) | TH-Cre | Jackson Laboratory Stock No: 008601 | RRID:IMSR_JAX:008601 | |
| Cell line (*human*) | HEK 293FT | Thermo Fisher Scientific | Catalog number: R70007 | |
| Chemical compound, drug | CNQX | Tocris Bioscience | Cat# 0190 | |
| Chemical compound, drug | AP5 | Tocris Bioscience | Cat# 0106 | |
| Chemical compound, drug | TTX | Tocris Bioscience | Cat# 1078 | |
| Chemical compound, drug | Fura-2 AM | Thermo Fisher Scientific | Cat# F1221 | |
| Antibody | Anti-FLAG (Mouse monoclone M2) | Sigma-Aldrich | Cat# F1804 RRID:AB_262044 | ICC (1:1000) |
| Antibody | Anti-TH (Rabbit polyclonal) | Abcam | Cat# ab6211 RRID:AB_2240393 | ICC (1:2000) |
| Antibody | Anti-synaptophysin (Rabbit polyclonal) | Aviva Systems Biology | Cat# ARP45435_P050 RRID: AB_2048301 | ICC (1:1000) |
| Antibody | Anti-synaptophysin (Mouse monoclonal) | Sysy | Cat# 101011 RRID: AB_887824 | ICC (1:1000) |
| Antibody | Anti-AP2 (Mouse monoclonal) | abcam | Cat# ab2730 RRID: AB_303255 | ICC (1:200) |
| Antibody | Aanti-Syt1 (Mouse monoclonal) | Sysy | Cat# 105011 RRID: AB_887832 | ICC (1:1000) |
| Antibody | Anti-clathrin (Mouse monoclonal) | Pierce | Cat# MA1-065 RRID: AB_2083179 | ICC (1:500) |
| Antibody | Anti-dynamin (Rabbit polyclonal) | Abcam | Cat# ab3456 RRID: AB_303818 | ICC (1:1000) |
| Antibody | Anti-mouse 555 (Donkey polyclonal) | Thermo Fisher Scientific | Cat# A-31570 RRID:AB_2536180 | ICC (1:1000) |
| Antibody | Anti-rabbit 488 (Donkey polyclonal) | Thermo Fisher Scientific | Cat# A-21206 RRID:AB_2535792 | ICC (1:1000) |
| Antibody | Anti-mouse 647 (Donkey polyclonal) | Thermo Fisher Scientific | Cat# A32787 RRID:AB_2762830 | ICC (1:1000) |
| Antibody | Anti-rabbit 555 (Goat polyclonal) | Thermo Fisher Scientific | Cat# A32732 RRID:AB_2633281 | ICC (1:1000) |
| Antibody | Anti-mouse 488 (Goat polyclonal) | Thermo Fisher Scientific | Cat# A32723 RRID:AB_2633275 | ICC (1:1000) |
| Antibody | Anti-rabbit 647 (Goat polyclonal) | Thermo Fisher Scientific | Cat# A32733TR RRID:AB_2866492 | ICC (1:1000) |
| Recombinant DNA reagent | pCDH-EF1-MCS-T2A-copGFP | System Biosciences | Cat# CD526A-1 | A lentiviral vector |
| Recombinant DNA reagent | CMV::SypHy 4 A | Dr Leon Lagnado PubMed 16982422 | Cat# 24478 RRID:Addgene_24478 | SypHy was subcloned into a pCDH lentiviral vector containing a synapsin 1 promoter in lab |
| Recombinant DNA reagent | pCDH-SYN1-sypHy | This paper | N/A | Plasmid was packaged into lentivirus, which was used to transduce primary neurons |

| Reagent type (species) or resource | Designation | Source or reference | Identifiers | Additional information |
|---|---|---|---|---|
| Recombinant DNA reagent | pCDH-EF1-sypHy-T2A-TRPM7$^{WT}$ | This paper | N/A | Plasmid was used to transfect TRPM7 KO neurons |
| Recombinant DNA reagent | pCDH-EF1-sypHy-T2A-TRPM7$^{LCF}$ | This paper | N/A | Plasmid was used to transfect TRPM7 KO neurons |
| Recombinant DNA reagent | pCDH-EF1- TRPM7$^{WT}$ (Flag tag on N-term) | This paper | N/A | Plasmid was packaged into lentivirus, which was then used to infect TRPM7 KO chromaffin cells |
| Recombinant DNA reagent | pCDH-EF1-TRPM7$^{LCF}$ (Flag tag on N-term) | This paper | N/A | Plasmid was packaged into lentivirus, which was then used to infect TRPM7 KO chromaffin cells |
| Recombinant DNA reagent | TRPM7 (Flag tag on N-term) | Dr AM Scharenberg PubMed11385574 | Cat# 45482, Addgene RRID:Addgene_45482 | DNA was subcloned into a pCDH lentiviral vector containing a EF1 promoter in lab |
| Recombinant DNA reagent | vGlut1-pHluorin | Dr V Haucke PMID:21808019 | N/A | DNA was subcloned into a pCDH lentiviral vector containing a synapsin one promoter in lab |
| Recombinant DNA reagent | CaMKIIα-iGluSnFR | Dr ER Chapman PMID:32515733 | N/A | S72A variant was used in this paper; DNA was subcloned into a pCDH lentiviral vector containing a synapsin 1 promoter in lab |
| Recombinant DNA reagent | Syn-GCaMP6f | Dr TA Ryan lab PMID:28162809 | N/A | DNA was subcloned into a pCDH lentiviral vector containing a synapsin 1 promoter in lab |
| Software, algorithm | Igor Pro 8 | WaveMetrics | RRID:SCR_000325 | |
| Software, algorithm | Excel | Microsoft | RRID:SCR_016137 | |
| Software, algorithm | Micro-Manager | ImageJ | RRID:SCR_000415 | |
| software, algorithm | FIJI | Fiji.sc | RRID:SCR_002285 | |
| Software, algorithm | Adobe Illustrator CS2 | Adobe | RRID:SCR_010279 | |
| Software, algorithm | Prism | GraphPad | RRID:SCR_002798 | |
| Software, algorithm | NIS-Elements | Nikon | RRID:SCR_014329 | |

## Generation of TRPM7 conditional KO mice and mouse breeding

All animal experimental studies were approved by the Animal Care and Use Committee of the University of Illinois at Chicago and conformed to the guidelines of the National Institutes of Health (animal protocol number 19–189). All mating cages were exposed to a 12 hr light/dark cycle with food and water provided ad libitum.

*Trpm7*$^{fl/fl}$ (018784, a kind gift of Dr Clapham lab [*Jin et al., 2008*]), *Nestin-Cre* (003771, a kind gift from Dr Schütz lab [*Tronche et al., 1999*]), and *Th-Cre* (008601, a kind gift from Dr Dawson lab [*Savitt et al., 2005*]) mice were obtained from Jackson lab. TRPM7$^{fl/fl}$ mice were crossed with *Th-Cre* or *Nestin-Cre* line to produce hemizygous *Th-Cre* or *Nestin-Cre* mice, which were heterozygous for the loxP targeted (floxed) *Trpm7* gene (*Th-Cre/Trpm7*$^{fl/+}$ or *Nestin-Cre/Trpm7*$^{fl/+}$), respectively. Male *Th-Cre/Trpm7*$^{fl/+}$mice were then backcrossed to female *Trpm7*$^{fl/fl}$ mice to generate mice with TRPM7 KO catecholaminergic cells including chromaffin cells (*Th-Cre/Trpm*$^{fl/fl}$). Similarly, *Nestin-Cre/Trpm7*$^{fl/+}$ mice were backcrossed to *Trpm7*$^{fl/fl}$ mice to generate brain-specific TRPM7 KO mice (*Nestin-Cre/Trpm7*$^{fl/fl}$). Controls were the littermates carrying a single floxed allele of *Trpm7* (*Trpm7*$^{fl/+}$).

Cre transgenes were genotyped using tail biopsy DNA samples with PCR primers 5'-GCGGTCTG GCAGTAAAAACTATC-3' and 5'-GTGAAACAGCATTGCTGTCACTT-3' to detect a 100 bp transgenic fragment. The TRPM7 floxed and WT alleles were detected using the primer set: forward, 5'- TTTC TCCAATTAGCCCTGTAGA-3'; reverse, 5'-CTTGCCATTTTACCCAAATC-3', the products were 300 bp for the floxed allele and 193 bp for the WT allele. RT-PCR of adrenal medulla and cerebral cortex mRNA were used to confirm the excision of loxP-flanked (floxed) sequences, indicated by a 454 bp cDNA fragment of wild-type TRPM7, and a 225 bp fragment from Cre recombinase-excised, as previously described (*Jin et al., 2008*).

## Cultures of chromaffin cells, neurons, and HEK 293 cells

Chromaffin cells in culture, prepared from adrenal glands of newborn pups with either sex as previously described (*Gong et al., 2005*; *Yao et al., 2013*; *Yao et al., 2012*), were maintained at 37 °C in 5 % $CO_2$ and used within 4 days for electrophysiology or immunostaining. Lentiviral infections with TRPM7$^{WT}$ or TRPM7$^{LCF}$ in TRPM7 KO cells were carried out at days in vitro (DIV) 0, and electrophysiological recordings or immunostaining staining was performed 72–96 hr after infection.

Neuronal culture from cortex of newborn pups with either sex was prepared as described previously (*Gong and De Camilli, 2008*). Neurons in culture were maintained at 37 °C in 5 % $CO_2$ and used at DIV 14–16 for electrophysiology. Neurons at DIV 5 were transfected with vGlut1-pHluorin, TRPM7$^{WT}$-sypHy, or TRPM7$^{LCF}$-sypHy plasmids using Lipofectamine LTX (Thermo Fisher Scientific) or infected with sypHy, SyGCaMP6f, or iGluSnFR lentivirus, and imaging experiments were carried out on neurons at DIV 16–20.

HEK 293 cells were cultured in DMEM supplemented with 10 % FBS, 0.1 mM MEM non-essential amino acids, 6 mM L-glutamine, 1 mM MEM sodium pyruvate, 1 % Pen-Strep, and 500 μg/ml Geneticin antibiotic. Cells, which typically reached 80 % confluence every 3 days, were treated with trypsin-EDTA and passaged with a 1:10 ratio for the general maintenance. Cells at low density were seeded onto 24-well culture plates and infected with lentivirus carrying FLAG-tagged TRPM7$^{WT}$ or TRPM7$^{LCF}$, and monoclonal cell lines were then generated by limiting dilution. After confirming the monoclonal cell lines expressing either TRPM7$^{WT}$ or TRPM7$^{LCF}$ by FLAG immunostaining, the monoclonal cell lines were maintained at 37 °C in a 5 % $CO_2$ incubator for 48–72 hr before being trypsinized and seeded to PDL-coated coverslip for whole-cell patch clamping recordings.

## Cell-attached capacitance recordings and analyses of exocytic fusion-pore and endocytic fission-pore

Cell-attached capacitance recordings of membrane capacitance and conductance were performed as described previously (*Yao et al., 2013*; *Yao et al., 2012*). Fire polished pipettes had a typical resistance of ~2 M in the bath solution. The bath solution contained 130 mM NaCl, 5 mM KCl, 2 mM $CaCl_2$, 1 mM $MgCl_2$, 10 mM HEPES-NaOH, and 10 mM glucose; the pH was adjusted to 7.3 with NaOH. The solution in the cell-attached pipette contained 40 mM NaCl, 100 mM TEACl, 5 mM KCl, 2 mM $CaCl_2$, 1 mM $MgCl_2$, and 10 mM HEPES-NaOH; the pH was adjusted to 7.3 with NaOH. As mentioned previously (*Yao et al., 2013*; *Yao et al., 2012*), capacitance steps and fission-pore durations were considered as reliably detected for step sizes > 0.2 fF, and smaller capacitance steps are not included in the analysis. The number of endocytic events per patch detected in the cell-attached recordings is counted as the total number of downward capacitance steps within the first 5 min of recordings (*Yao et al., 2013*; *Yao et al., 2012*).

The fusion-pore conductance (Gp) for exocytic events or the fission-pore Gp for endocytic events can be calculated from both Re and Im traces as $Gp_{(Im+Re)} = \frac{Re^2 + Im^2}{Re}$ or from Im trace alone as $Gp_{(Im)} = \frac{\omega \cdot Cv}{\sqrt{(\frac{\omega \cdot Cv}{Im}) - 1}}$, as described (*Gong et al., 2007*; *Yao et al., 2013*; *Yao et al., 2012*). Analysis of fission-pore kinetics were restricted to fission-pores with durations > 15 ms, since shorter events were distorted by the lock-in amplifier low-pass filter (set to 1 ms, 24 dB) (*Yao et al., 2013*; *Yao et al., 2012*).

## Double (whole-cell/cell-attached) patch recordings in chromaffin cells

Double (cell-attached/whole-cell) patch recordings were performed to analyze endocytosis-associated current drift, in which the cell-attached patch pipette was utilized to simultaneously monitor endocytic events and the ionic current across the patch membrane. With the cell constantly clamped at –60 mV by the whole-cell patch pipette via an EPC-10 amplifier (HEKA Elektronik, Germany), the patch membrane was held at different voltages by varying the voltage in the cell-attached pipette. The solution in the whole-cell pipette contained 20 mM CsCl, 90 mM KCl, 20 TEACl, 10 mM HEPES-KOH, 2 mM EGTA, 2 mM $Na_2ATP$, and 0.3 $Na_3GTP$; the pH was adjusted to 7.3 with KOH with the osmolarity of 290 mmol/kg. The basal $Ca^{2+}$ level in the whole-cell pipette solution was buffered by 1.1 mM $CaCl_2$ and 2 mM EGTA to give a final free $Ca^{2+}$ level of ~150 nM. All the settings for cell-attached recordings and the solution in the cell-attached pipette were identical to descriptions in *Cell-attached capacitance recordings and analyses of exocytic fusion-pore and endocytic fission-pore*. Ionic currents from

the cell-attached pipette were low-pass filtered at 500 Hz. As illustrated in *Figure 1F*, the endocytosis-associated current drift was calculated as the difference between linear fittings of a 20–100 ms duration in the $I_{patch}$ trace before and after the capacitance drop due to endocytosis.

## Whole-cell capacitance recordings in chromaffin Cells

Whole-cell patch recordings, with pipettes of 2–4 MΩ resistance, on chromaffin cells were performed using an EPC-10 amplifier together with PULSE software (HEKA Electronics, Lambrecht/Pfalz, Germany). The external bathing solution contained 140 mM NaCl, 2.8 mM KCl, 10 mM $CaCl_2$, 1 mM $MgCl_2$, 10 mM HEPES, and 5 mM glucose (pH 7.25). The pipette solution contained 135 mM Cs-Glu, 2.8 mM KCl, 1 mM $MgCl_2$, 10 mM HEEPES, 2 mM MgATP, and 0.3 $Na_3GTP$ (pH 7.25). Basal $Ca^{2+}$ concentration in the pipette solution was buffered by a combination of 2 mM $CaCl_2$ and 5 mM EGTA to give a final free $Ca^{2+}$ concentration of ~200 nM. Capacitance measurements were performed using the software lock-in module of PULSE. A 1 kHz, 40 mV peak-to-peak sinusoid stimulus was applied on a DC potential of –90 mV. The resulting current was processed by using the Lindau–Neher technique (*Lindau and Neher, 1988*) to give estimates of the equivalent circuit parameters (Cm, Gm, and Gs). The reversal potential of the measured DC current was assumed to be 0 mV. ΔCm values were analyzed with Igor software (RRID: SCR_000325, WaveMetrics, Lake Oswego, OR). Exocytosis was estimated by using four 100 ms depolarizations to 0 mV (*Rettig and Neher, 2002*). Cm was averaged over a 50 ms pre-depolarization segment to give a baseline value. Our previous study has suggested that the transient gating capacitance artifact was negligible 40 ms after the end of a depolarization (*Gong et al., 2005*), thus, Cm was averaged over the 40–100 ms interval after depolarizations, and the baseline Cm value was subtracted to obtain the amount of exocytosis. Evoked $Ca^{2+}$ currents were measured under conditions in which most of the $K^+$ currents were blocked by intracellular $Cs^+$. Tetrodotoxin was not used to block $Na^+$ conductance in these experiments because it has been shown to prolong non-secretory capacitance transients in chromaffin cells. Instead, the first 10 ms of evoked inward current was excluded for estimations of $Ca^{2+}$ influx. No correction for leak currents was made and cells with a leak of >150 pS were discarded.

## Paired whole-cell recordings of evoked IPSCs in neurons

Paired whole-cell patch-clamp recordings were performed on DIV 14–16 neurons, continuously perfused with an extracellular solution containing 130 mM NaCl, 3 mM KCl, 10 mM HEPES-NaOH, 2 mM $CaCl_2$, 1 mM $MgCl_2$, and 10 mM glucose, pH adjusted to 7.3 with NaOH. Evoked IPSCs were isolated by adding amino-5-phosphonopentanoate (AP5, Tocris) at 50 µM and 6-cyano-7-nitroquinoxaline-2,3-dione (CNQX, Tocris) at 20 µM in the bath solution to block excitatory synaptic transmission. The pipette solution for presynaptic neurons contained 125 mM KCl, 10 mM HEPES, 2 mM EGTA-KOH, 1 mM $CaCl_2$, 1 mM $MgCl_2$, 2 mM $Na_2ATP$, 0.4 mM $Na_3GTP$, and 5 mM phosphocreatine disodium salt, the pH adjusted to 7.3 with KOH. For postsynaptic neurons, the pipette solution contained 125 mM CsCl, 10 mM HEPES, 2 mM EGTA-CsOH, 1 mM $CaCl_2$, 1 mM $MgCl_2$, 2 mM $Na_2ATP$, 0.4 mM $Na_3GTP$, 5 mM QX-314, and 5 mM phosphocreatine disodium salt, pH adjusted to 7.3 with CsOH.

IPSCs were recorded at –70 mV on pyramidal neurons using dual whole-cell configurations by evoking a nearby inhibitory neuron with a 1 ms depolarization from –70 to +30 mV. Recordings were acquired with an EPC-10 patch-clamp amplifier for the presynaptic voltage clamping and an EPC-7 plus patch-clamp amplifier for postsynaptic recordings. Data were filtered at 1 kHz and digitized at 5 kHz. Patch pipettes had a typical resistance of 2–3 MΩ. Criteria for inclusion required that Rs remained below 10 MΩ and did not increase by more than 20 % during recordings.

Analysis was carried out with Igor software, using custom-written analysis procedures. Peak amplitudes of IPSCs were extracted by subtracting the extrapolated residual current caused by previous stimulation. The amplitude of IPSCs during train stimulations was normalized by dividing the peak of each response to the peak amplitude of the first response.

## Whole-cell recordings of TRPM7 currents in HEK 293 cells

TRPM7 currents were monitored in HEK 293 cells using the whole-cell configuration as previously reported (*Aarts et al., 2003*). The bath solution contained 130 mM NaCl, 5 mM KCl, 1 mM $CaCl_2$, 1 mM $MgCl_2$, 10 mM HEPES-NaOH, and 10 mM glucose, 0.1 mM $CdCl_2$, 1 µM TTX; the pH was adjusted to 7.3 with NaOH, and the osmolarity was ~310 mmol/kg. The solution in the whole-cell

pipette contained 120 mM Cs-MeSO$_3$, 8 mM NaCl, 2 mM CaCl$_2$, 10 mM EGTA-CsOH, and 10 mM HEPES; the pH was adjusted to 7.3 with CsOH, and the osmolarity was adjusted to 290 mmol/kg. The voltage ramp protocol holds cells at 0 mV, steps to –100 mV for 40 ms, and then ramps to +100 mV over 500 ms, holding at +100 mV for 40 ms before stepping back to 0 mV, and this protocol is repeated every 5 s during recording. Series resistance was monitored by applying a voltage step of 10 mV. Current signals were filtered at 3 kHz and digitized at 5 kHz. Only recordings with a pipette-membrane seal resistance >2 GΩ were included. Current traces and time courses were analyzed with customized macro for Igor software.

## Carbon fiber amperometry

Chromaffin cells were incubated in the bath solution containing 140 mM NaCl, 5 mM KCl, 2 mM CaCl$_2$, 1 mM MgCl$_2$, 10 mM HEPES-NaOH, and 10 mM glucose, pH 7.3 with NaOH. The stimulation solution contained 45 mM NaCl, 90 mM KCl, 2 mM CaCl$_2$, 1 mM MgCl$_2$, 10 mM HEPES-NaOH, and 10 mM glucose. Conventional carbon fiber amperometry for catecholamine detection used 5 µm carbon fibers (ALA Scientific, Farmingdale, NY) as described previously (*Gong et al., 2007*; *Gong et al., 2005*; *Yao et al., 2012*). The freshly cut tip of the carbon fiber electrode was positioned closely against the cell surface to minimize the diffusion distance from release sites, and cells were stimulated with 90 mM KCl solution for 5 s by a pressurized perfusion system (VC3, ALA Scientific) placed ~40 µm away from the cell. The amperometric current, generated by catecholamine oxidation at the exposed tip of the carbon fiber electrode due to stimulation, was measured using an EPC-7 plus amplifier at a holding potential of +700 mV. Amperometric signals were low-pass filtered at 3 kHz and digitized at 5 kHz. Amperometric recordings were collected and then analyzed with a customized macro for Igor software to extract spike information according to the criteria of *Chow et al., 1992*. The following criteria were set in single-spike analysis: (i) only spikes > 10 pA were considered; (ii) the maximum number of spikes analyzed per cell was set to 100; (iii) foot duration was delimited by the baseline and spike onset as shown previously (*Chow et al., 1992*), and (iv) spikes with a foot duration of <0.5 ms were excluded for the assays. Data are presented as mean ± SEM of the cell medians. Thus, the numbers used for statistical tests are the number of cells. The number of amperometric spikes was counted as the total number of spikes with an amplitude >10 pA within 15 s after stimulation.

## pHluorin-, SyGCaMP6f- and iGluSnFR-based live-cell imaging in neurons

Neurons on coverslips were continuously perfused at a flow rate of ~1 ml/min with bath solution (130 mM NaCl, 2.8 mM KCl, 2 mM CaCl$_2$, 1 mM MgCl$_2$, 10 mM glucose, 10 mM HEPES [pH 7.4]; ~310 mOsm). All imaging experiments were performed on an Olympus IX51 microscope with a 60 × oil immersion objective at room temperature (25°C ± 1 °C), with a GFP-3035B filter cube (472/30 nm excitation, 520/35 nm emission, 495 nm dichroic mirror) (Semrock, Rochester, NY) as the filter set. To elicit responses, APs were delivered by electric field stimulation at 10 V/cm with 1 ms duration via two parallel platinum wires embedded within the perfusion chamber with 7 mm spacing; 20 µM CNQX (Tocris Bioscience) and 50 µM D,L-2-amino-5-phosphonovaleric acid (AP5, Tocris Bioscience) were included in the bath solution to prevent recurrent activities.

For train stimulation experiments using sypHy, vGlut1-pHluorin, or SyGCaMP6f, a 120 W mercury vapor short Arc lamp (X-cite, 120PC, Exfo) was utilized as the light source for excitation. Images were captured by a Photometrics HQ2 coolSNAP CCD camera controlled by µManager software ( micro-manager.org) at a 1–2 s interval with 100 or 200 ms exposure time at 25 % illumination. To measure reacidification rate of newly formed endocytic vesicles using sypHy as described previously (*Atluri and Ryan, 2006*), neurons on coverslip were rapidly superfused with a valve controlled pressurized perfusion system (ALA Scientific, Farmingdale, NY). The perfusing buffer was switched between the standard pH 7.3 bath solution described above and pH 5.5 buffer with the HEPES substituted with equi-molar MES. The electrical stimulation was thus time-locked to frame acquisition and buffer changes.

For single AP stimulation using SyGCaMP6f and train stimulation using iGluSnFR, Lambda 421 Optical Beam Combining System (Sutter instrument, Novato, CA) was used as the light source for illumination. To achieve high quantum efficiency, we used the Prime BSI Scientific CMOS (sCMOS) camera (Teledyne Photometrics). With 20 ms exposure time, the illumination via Lambda 421 and data

acquisition via camera were controlled by Igor software to achieve a sampling rate of 50 Hz. The light intensity of illumination was set at 10 % to minimize photobleaching.

## Fura-2 AM-based Ca²⁺ imaging in chromaffin cells and neurons

Chromaffin cells or neurons in culture were loaded with freshly thawed, cell permeant Fura-2 AM (Thermo Fisher Scientific, Waltham, MA) at the final concentration of 5 μM for 30 min at room temperature (*Jiang et al., 2019*). Cells or neurons were then gently rinsed three times with the regular extracellular solution of chromaffin cells or neurons to remove the excess dye and recovered for 20 min in fresh bath solution before experiments.

Chromaffin cells or neurons loaded with Fura-2 AM were imaged with a Prime 95B sCMOS camera mounted onto a Nikon Eclipse Ti2 inverted microscope using a 40×/1.30 oil-immersion objective with 1.5 × external magnification. Fluorescence emission images (525 nm) of chromaffin cells or neurons were acquired for 10 s continuously in response to alternating 340 and 380 nm excitation light delivered by a Retra Light Engine with 50 ms exposure. Both diodes were co-illuminated and transmitted to the same optical path through a dichroic beam combiner. Presynaptic terminals in neurons were identified by lentivirus-mediated expressions of vGlut1-pHluorin. Imaging acquisition was performed using NIS-elements software (RRIS: SCR_014329).

## Immunofluorescence and confocal imaging of chromaffin cells and neurons

Chromaffin cells or neurons in culture were fixed with 4 % paraformaldehyde for 10 min at room temperature. After permeabilization in blocking PBS buffer containing 3 % BSA, 5 % normal donkey serum, and 0.1 % Triton X-100 for 15 min, the fixed cells or neurons were incubated with primary antibodies overnight at 4 °C, followed by fluorescent dye-conjugated secondary antibodies at 1:1000 dilution for 1 hr at room temperature.

Confocal fluorescence images of chromaffin cells or neurons were captured using an Olympus Fluoview FV10i laser-scanning confocal microscope with a 60× oil-immersion objective (NA 1.4) and 473 , 559 , and 635 nm lasers. The pinhole was set at 50 μm for imaging capturing, and the image size was 1024 × 1024. For comparisons, the settings of the laser intensity and sensitivity of each channel are identical during image capture between WT and TRPM7 KO cells for each individual protein we tested. For different protein we tested, however, the laser intensity and sensitivity of each channel were adjusted differently to minimize photobleaching and saturation of fluorophores.

## Image analysis

Images were analyzed in ImageJ (RRID: SCR_000415, http://rsb.info.nih.gov/ij) using custom-written plugins (http://rsb.info.nih.gov/ij/plugins/time-series.html) and Igor software using custom-written procedures. Photobleaching was less than 2 % for all images included within analysis and thus was not corrected. To analyze images from sypHy, vGlut1-pHluorin, or SyGCaMP6f experiments, regions of interest (ROIs) of identical size (4 × 4 pixels) were placed in the center of individual synaptic boutons reacting to stimuli, and fluorescence changes were tracked throughout the image stack. In general, 10–20 ROIs were analyzed from each coverslip for sypHy or vGlut1-pHluorin experiments, and 20–40 ROIs for all SyGCaMP6f experiments. For iGluSnFR analysis, ROIs were created as previously reported (*Vevea and Chapman, 2020*). Briefly, ROIs (>10 pixels), defined by a series of image subtractions and thresholding (*Vevea and Chapman, 2020*), were used to measure fluorescence changes of image stacks.

Fluorescent intensities of individual ROIs prior to stimuli were averaged as baseline (F0). The fluorescent changes (ΔF) were normalized to the peak of fluorescent increase (ΔFmax) as ΔF/ΔFmax for sypHy, vGlut1-pHluorin, or iGluSnFR experiments and as ΔF/F0 for SyGCaMP6f experiments. Normalized ΔF of individual ROIs were taken as independent replicates (n) for sypHy and vGlut1-pHluorin experiments with the average of normalized ΔF of ROIs for each coverslip as the independent replicates (n) for SyGCaMP6f and iGluSnFR experiments.

For analysis of Fura-2 AM data, each individual chromaffin cell was defined as a round ROI, while line ROIs with two-pixel thickness were drawn across presynaptic terminals identified by vGlut1-pHluorin in neurons. The fluorescence emission intensity ratio corresponding to 340 nm/380 nm excitation was obtained for chromaffin cells and presynaptic terminals in neurons.

For quantifications of expression levels of endocytic proteins from confocal images, maximum intensity Z-series stacks were created in the image analysis program ImageJ (imagej.nih.gov/ij). The whole cell was selected as ROI for chromaffin cells and ROI of presynaptic terminals, identified by synaptophysin immunostaining, were defined as a 4 × 4 pixel area in neurons. The fluorescent intensity of ROI was subtracted by fluorescent intensity of a background ROI with the same size.

## DNA constructs

GFP in the lentiviral vector (pCDH-EF1-MCS-T2A-GFP) was removed and replaced with an artificial DNA fragment containing *Pme* I and *Apa* I restriction sites using sense oligo: 5'-CCGGAATGTTTA AACGGGCCCG and antisense oligo: 5'-TCGACGGGCCCGTTTAAACATT-3'. To engineer pCDH-EF1-sypHy, sypHy (#24478 of Addgene, a kind gift from Dr L Lagnado [*Granseth et al., 2006*]) was inserted into MCS by adding an *Xba* I site to the 5' end (primer: 5'-CTTTTCTAGAATGGACGTGGTGAATCA-3') and a *Not* I site to the 3' end (primer: 5'-CAGCGGCCGCCATCTGATTGGAGAAGGAG-3'). The EF1 promoter was then replaced with a human synapsin 1 (SYN1) promoter to produce pCDH-SYN1-sypHy, which was packaged into lentivirus and used for neuronal infections.

SyGCaMP6, a kind gift from Dr Timothy A Ryan's lab, was generated by adding GCaMP6f to the C-terminus of the mouse sequence of synaptophysin as previously reported (*Kim and Ryan, 2013*). SyGCaMP6f was first cloned by the standard PCR protocol using Phusion DNA polymerase (forward primer: 5'-ACTCGGATCCCCTCTAGAATGGACGTGGTGAATCAGCTG-3'; reverse primer: 5'- GATT GATATCTCACTTCGCTGTCATCATTTG-3'). The PCR fragment was then enzyme-digested and ligated between the *BamHI* and *EcoRV* sites to obtain pCDH-SYN1- SyGCaMP6f by replacing sypHy in the pCDH-SYN1-sypHy vector.

Modified iGluSnFR (*Marvin et al., 2018*), with the final four amino acid removed and a Golgi export sequence and ER exit motif included, is a kind gift from Dr Edwin Chapman's lab (*Bradberry et al., 2020*). iGluSnFR was subcloned and inserted between the BamH1 and Not1 restriction sites in the lentiviral vector (pCDH-CaMKIIα -MCS, a construct we created in lab) to obtain pCDH-CaMKIIα-iGluSnFR (forward primer: 5'- ATAAGGATCCATGGAGACAGACACACTCCTGCTATG-3', and reverse primer: 5'- CGATGCGGCCGCTTAGAGGGCAACTTCATTTTCATAGC-3').

To express TRPM7 in KO chromaffin cells with lentivirus, TRPM7 (#45482 of Addgene, a kind gift of Dr A Scharenberg's lab [*Nadler et al., 2001*]) was subcloned into a lentiviral vector via *Pme* I and *Apa* I restriction sites to generate pCDH-EF1-TRPM7$^{WT}$. To engineer TRPM7 mutant with a loss of channel function (TRPM7$^{LCF}$, amino acid 1090–1092, NLL > FAP) (*Krapivinsky et al., 2006*), site-directed mutagenesis was performed using *Pfu* hotstart DNA polymerase (forward primer: 5'-CAGT ATATCATTATGGTTTTCGCTCCTATCGCATTTTTCAATAAT-3', and reverse primer: 5'-ATTATTGAAAAA TGCGATAGGAGCGAAAACCATAAT GATATACTG-3'). Meanwhile, to express TRPM7 in KO neurons, TRPM7$^{WT}$ or TRPM7$^{LCF}$ was digested out with *Pme* I and *Apa* I restriction sites and ligated into pCDH-EF1-sypHy with the artificial DNA fragment to produce pCDH-EF1-sypHy-T2A-TRPM7.

## Lentiviral productions

Low-passage HEK 293 cells (Thermo Fisher Scientific) were co-transfected with 10 µg lentiviral vector (pCDH-SYN1-sypHy, pCDH-SYN1-SyGCaMP6f, pCDH-CaMKIIα-iGluSnFR, pCDH-EF1-TRPM7$^{WT}$, or pCDH-EF1-TRPM7$^{LCF}$) along with 7 µg packaging vector psPAX2 and 3 µg envelope vector pMD2. VSVG using the polyethylenimine-mediated transfection method. The supernatant containing lentiviral particles was collected and filtered through 0.45 µm filter to remove cell debris at 48 and 72 hr. Lentiviral particles, concentrated by PEG-it precipitation kits from System Bioscience (Palo Alto, CA), were re-suspended in cold PBS and stored at –80 °C.

There is a logarithmic inverse relationship between packaging size and lentiviral titer, that is, the titer drops by one log for every 2 kb of insert (*Kumar et al., 2001*). Since *Trpm7* is a relatively large gene of >5.5 kb, to increase the packaging efficiency, *Flag* was tagged to the 5'-end of *Trpm7* variants for lentivirus, which was utilized to infect KO chromaffin cells. Our experiments showed that the majority of chromaffin cells were infected by anti-FLAG immunostainings (94% ± 1.4% from three independent experiments).

## Statistical analysis

Data were tested for normal distribution and, if necessary, log-transformed for the fission-pore duration in *Figures 1E and 4G*, and *Figure 1—figure supplement 2B* to fulfill the criteria of normal distributions

for Student's t-tests. All statistical analyses were performed with Prism (RRID: SCR_002798, GraphPad, San Diego, CA).

No statistical method was used to predetermine sample size, but our samples sizes are similar to those reported in previous studies (*Fernández-Alfonso and Ryan, 2004*; *Granseth et al., 2006*; *Soykan et al., 2017*; *Yao et al., 2013*; *Yao et al., 2012*). Data was expressed as mean ± SEM, animals or cells were randomly assigned into control or experimental groups. Statistical analysis was performed with unpaired two-tailed Student's t-test except paired two-tailed Student's t-test in *Figure 1—figure supplement 4B* and Newman of one-way ANOVA in *Figure 5B* and *Figure 4— figure supplement 2B*.

## Acknowledgements

This work is supported by NIH (R01NS110533) to L-WG and NSF (IOS-1346826) to BSG. The authors are grateful to Dr V Haucke for vGlut1-pHluorin, Dr TA Ryan for SyGCaMP6f, Dr ER Chapman for iGluSnFR, and Dr AM Scharenberg for TRPM7 plasmid (#45482, Addgene).

## Additional information

### Funding

| Funder | Grant reference number | Author |
|---|---|---|
| NIH Office of the Director | R01NS110533 | Liang-Wei Gong |
| NSF | IOS-1346826 | Brian S Grewe |

The funders had no role in study design, data collection and interpretation, or the decision to submit the work for publication.

### Author contributions

Zhong-Jiao Jiang, Conceptualization, Formal analysis, Investigation, Methodology, Software, Validation, Writing – original draft, Writing – review and editing; Wenping Li, Conceptualization, Formal analysis, Investigation, Methodology, Writing – original draft, Writing – review and editing; Li-Hua Yao, Kelly Varga, Simon Alford, Investigation; Badeia Saed, Brian S Grewe, Andrea McGinley, Investigation, Methodology; Yan Rao, Formal analysis, Investigation, Writing – review and editing; Ying S Hu, Investigation, Methodology, Resources; Liang-Wei Gong, Conceptualization, Funding acquisition, Project administration, Supervision, Writing – original draft, Writing – review and editing

### Author ORCIDs

Brian S Grewe http://orcid.org/0000-0002-8287-6588
Liang-Wei Gong http://orcid.org/0000-0002-5868-6039

### Ethics

All animal experimental studies were approved by the Animal Care and Use Committee of the University of Illinois at Chicago and conformed to the guidelines of the National Institutes of Health.(animal protocol number 19-189).

### Decision letter and Author response

Decision letter https://doi.org/10.7554/eLife.66709.sa1
Author response https://doi.org/10.7554/eLife.66709.sa2

## Additional files

### Supplementary files
• Transparent reporting form

### Data availability
All data generated or analyzed during this study are included in the manuscript and supporting files.

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
