## [Decision Letter]

**Acceptance summary:**

Your paper describes an interesting new role of the ion channel TRPM7 in the regulated release of neurotransmitters. The corresponding data provide important new insights into the mechanisms by which synaptic transmitter release is fine-tuned.

**Decision letter after peer review:**

Thank you for submitting your article "TRPM7 is critical for short-term synaptic depression by regulating synaptic vesicle endocytosis" for consideration by *eLife*. Your article has been reviewed by 3 peer reviewers, one of whom is a member of our Board of Reviewing Editors, and the evaluation has been overseen by Lu Chen as the Senior Editor. The reviewers have opted to remain anonymous.

Essential revisions:

Essential – Not Requiring Additional Experimental Work

1. The TRPM7 KO strategy is based on the deletion of exon 17. The initial description of the KO (Jin et al., 2008) does not clearly show whether there is a possibility of the generation of a truncated variant of the channel or one with a deletion within the sequence upstream of the TMDs. Can this been excluded?

2. For all data in the manuscript, the authors need to show data points, not just bar graphs with mean and SE.

3. The data in Figure 1E showing longer fission pore durations in the KO are interesting (and similarly, the data in Figure 4H on the rescue), but in the legend it becomes clear that only 42 events in each group were considered, which yielded p<0.05. Strikingly, though, the authors measured from no less than 83 and 110 cells with approximately 6 events per cell (Figure 1B), so overall, <10% of the events were included in panel E. Based on the Methods, the authors only considered events with step sizes >0.2 fF and durations >15 ms, citing a low-pass filter set at 1 ms (24 dB). The reviewers understand that the method has limitations (and the authors are clearly competent in the field), but it is a concern to be concluding based on such a small selection of events. With a filter setting at 1 ms, could the authors attempt to visualize the entire distribution of events down to 5 ms, to inspect if the difference between WT and KO is found only for the longest events, or independently of the event duration?

4. In the example of the current flowing through the patch of a WT but not KO cell in Figure 1F, the capacitance change appears to be >10 times in size compared to the example shown in Figure 1A. If this is not a simple typo, an explanation is required as to whether this finding even applies to the smaller events.

5. The authors aimed to study the physiological function of presynaptic TRPM7, and they conclude that TRPM7 might act as a Ca^2+^-influx pathway. However, the pipette-solutions for whole-cell experiments (in chromaffin cells and HEK-cells, where TRPM7-related currents were apparently detected directly, in Figure 1F and Figure 4A) did not contain Mg^2+^ (in chromaffin cells Na-ATP and Na-GTP were used; in HEK-cells no nucleotides or magnesium were included), and TRPM7 is described as a ion channel that is fully inhibited by 2 mM intracellular Mg-ATP (Nadler et al., 2001). Therefore, it is not straight forward to see how Ca^2+^ would permeate the channel under physiological conditions. The authors have data (e.g. CGaMP6) consistent with Ca^2+^-influx, and they show that a mutation (LCF) blocking permeation also blocks function in chromaffin cells, but this is not conclusive evidence that permeation is involved. The authors need to discuss the negative regulation of TRPM7 by Mg-ATP carefully (it seems to not even be mentioned), and whether physiologically relevant conditions are likely to arise where this block would be relieved. They should also discuss their choice of pipette solutions. Finally they should discuss whether their data might be consistent with other scenarios than Ca^2+^-influx through TRPM7.

6. It is unclear why the KO-induced changes in vesicle fusion do not show-up as a slower rise in the SypHy measurements shown in Figure 3. This should be briefly discussed.

7. The explanatory connection that the authors make between the TRPM7-KO effects on calcium signals, endocytosis, and short-term plasticity seems forced. The KO reduces presynaptic calcium signals (likely also with endocytosis during stimulation trains) and increases synaptic depression already very early in the train. There are more parsimonious explanations for the short-term plasticity change: Reduced bulk calcium levels could cause reduced calcium-dependent activation of the vesicle priming machinery or even reduced activation of calcium sensors in fusion (Synaptotagmins). This aspect should be considered in the discussion part of the paper. It may well be that the short-term plasticity defect in the TRPM7 KO neurons is not directly related to the endocytosis defect. Further, it seems hard to imagine that a selective slow-down of endocytosis would become manifest as an appreciable change in exocytosis already after only 10 APs – unless one considers 'site clearing' as proposed by Neher and Sakaba.

8. It is unclear why the 'endocytosis-associated conductance' the authors describe was missed in the many studies employing whole-cell recordings in chromaffin cells. Do the authors assume that detection failed in these cases due to the outward-rectifying character and/or the fact that typical measurements are done at -80 mV? This issue should be discussed.

9. The time scale of changes in endocytosis kinetics in TRPM7 KO chromaffin granules is of no known physiological impact. This should be discussed.

Essential – Requiring Additional Experiments

1. A basic analysis of the morphology of TRPM7 KO chromaffin cells and neurons is needed to complement the present data. Further, it should be tested if the TRPM7 KO affects the expression or localization of proteins that are critically involved in presynapse function, especially in endocytosis and endocytotic fission (e.g. Dynamins).

2. It should be properly verified that WT and mutant versions of TRPM7 are expressed and targeted in the same fashion. Figure S5 does not support the case that the mutant protein is correctly trafficked. It is unclear what the anti-FLAG staining corresponds to in the KO cells infected with a virus with an empty vector. This staining appears to be vesicular, which is also present in the TRPM7-WT infected cells, but not in the LCF infection. A higher resolution approach is required to localize the FLAG staining (confocal might be sufficient). If TRPM7 is no longer targeted correctly when it has the conduction mutation, one cannot conclude that the current through the channel is what is important.

3. Given the ability of the authors to perform very high-end electrophysiological analyses of transmitter secretion from chromaffin cells, the use of KCl^-^depolarization to assess TRPM7-KO-induced changes in exocytosis from chromaffin cells appears 'crude'. This type of stimulation is massive and somewhat unphysiological. The key question that remains open is whether exocytosis triggered by more physiological stimuli is affected by the KO (e.g. as assessed by short depolarisation stimuli combined with capacitance measurements). This should be tested in a small KO-vs.WT analysis.

4. Figure 3E-F show reacidification in WT and KO neurons. The description of this experiment is insufficient. The rationale of the experiment is a bit diffuse and it is unclear how the quantification (i.e. on what part of the trace). Moreover, the trace in panel E seems to be inconsistent with the quantification in panel F. In panel E, the reduction in DeltaF in the presence of the second low-pH pulse is steeper in the WT than in the KO. Can this be explained? In essence, these data in their present state are not sufficient to draw unequivocal conclusions regarding reacidification. The authors should include controls with bafilomycin, to understand whether the slopes they measure are actually reporting reacidification.

5. Reacidification kinetics are not resolved, yet are reported as a single parameter of an exponential fit. Furthermore, when the acid was removed, the KO cells should have had higher fluorescence, since it is decaying more slowly (e.g. difference WT vs. KO at the 80 s time point). The acid quenching experiment showed no difference. Given that the authors are not resolving an exponential decay of the acid-quenched surface signal, it would be more appropriate to simply report what fraction of the signal early on after the end of the stimulus – where there is a real difference in the amount of the signal that has decayed (e.g. at the 60 s time point) – can be quenched by acid perfusion. If it is the same fraction for WT and KO, the differences are not explained by the build-up of an alkaline pool, which is really what the authors want to assess here.

6. It is important to know if the phenomena described in the present study occur under physiological conditions (i.e. 37{degree sign}C, ~1.25 mM Ca). The temperature in particular is important as cooling can substantially distort the relevance of different molecular processes. The reviewers do not expect a repetition of all experiments at 37{degree sign}C. The easiest might be an experiment on pHluorin assessment of vesicle recycling in neurons, showing that at 37{degree sign}C there is a measurable impact on the pHluorin recovery kinetics due to the absence or presence of external Ca^2+^.

[Editors' note: further revisions were suggested prior to acceptance, as described below.]

Thank you for resubmitting your work entitled "TRPM7 is critical for short-term synaptic depression by regulating synaptic vesicle endocytosis" for further consideration by *eLife*. Your revised article has been reviewed by 3 peer reviewers, one of whom is a member of our Board of Reviewing Editors, and the evaluation has been overseen by Lu Chen as the Senior Editor.

Your manuscript has been very substantially improved. However, the reviewers have identified three issues that need to be resolved before the paper can be published in *eLife*:

1. The new confocal images added to Figure 3 (supplement 3, panels A-D; supplement 4, panels A, C, E, G) and to Figure 4 (supplement 2, panel C) are of very poor quality (very low resolution, blurred, pixelated) and do not look like confocal images (rather like ones that have been improperly digitally compressed). In fact, the new panel C in Figure 4, supplement 2 appears to be of even less resolution than the original images in panel A. In essence, the problematic panels indicated above are not of publication quality and need to be redone.

2. The description of confocal microscopy in the methods part is insufficient and requires more detail (i.e. numerical aperture and type of objective, light source, pinhole setting, etc.).

3. Regarding Figure 5, supplement 1, it seems that the 340 nm images in panel A are saturated. If so, it would be impossible to do a ratio analysis as shown in panel B. This issue needs to be explained and resolved.

---

## [Author Response]

Essential revisions:Essential – Not Requiring Additional Experimental Work1. The TRPM7 KO strategy is based on the deletion of exon 17. The initial description of the KO (Jin et al., 2008) does not clearly show whether there is a possibility of the generation of a truncated variant of the channel or one with a deletion within the sequence upstream of the TMDs. Can this been excluded?

We appreciate reviewer’s comment. As described in supplementary text in the paper by Jin et al. (Jin et al., 2008), the Cre-mediated deletion of exon 17, which encodes the protein sequence preceding the first transmembrane segment of TRPM7, results in a frame shift that disrupts the translation of the downstream sequence, thus generating TRPM7 with only N-terminal lacking all 7 transmembrane domains and the c-terminal tail. Therefore, the possibility of the generation of a truncated variant of the channel can be excluded.

2. For all data in the manuscript, the authors need to show data points, not just bar graphs with mean and SE.

Following the reviewer’s comment, all data in the figures are presented as mean±SE with all original data points except for panels Figure 1G and Figure 3—figure supplement 5C. And the original data points for these two panels are provided in the related source data files in the revised manuscript.

3. The data in Figure 1E showing longer fission pore durations in the KO are interesting (and similarly, the data in Figure 4H on the rescue), but in the legend it becomes clear that only 42 events in each group were considered, which yielded p<0.05. Strikingly, though, the authors measured from no less than 83 and 110 cells with approximately 6 events per cell (Figure 1B), so overall, <10% of the events were included in panel E. Based on the Methods, the authors only considered events with step sizes >0.2 fF and durations >15 ms, citing a low-pass filter set at 1 ms (24 dB). The reviewers understand that the method has limitations (and the authors are clearly competent in the field), but it is a concern to be concluding based on such a small selection of events. With a filter setting at 1 ms, could the authors attempt to visualize the entire distribution of events down to 5 ms, to inspect if the difference between WT and KO is found only for the longest events, or independently of the event duration?

To answer the reviewer’s question, we have performed a re-analysis of this data set. Results from this re-analysis showed that the percentage of endocytic events in different duration ranges (<5 ms; between 5 ms and 15 ms; >15 ms) was comparable between WT and KO cells. Consistent with an increase in the fission-pore duration in KO cells with inclusion of >15 ms events as shown in Figure 1E in the initial analysis, our re-analysis also showed an increase in the fission-pore duration in TRPM7 KO cells with additional inclusions of >5 ms but <15ms events, which is presented now in Figure 1—figure supplement 2 in the revised manuscript. It is thus likely that the difference in the fission-pore duration between WT and KO cells is independent of event duration. In addition, comparisons between WT and KO cells in the distribution of endocytic events with their fission-pore duration >5 ms or >15 ms are presented in this Figure 1—figure supplement 2 (>15 ms: panel C; >5 ms: panel D). We thank the reviewer for this comment, which prompt us to re-analyze this set of data, making our conclusion on the increase in the fission-pore duration in TRPM7 KO cells more convincing.

4. In the example of the current flowing through the patch of a WT but not KO cell in Figure 1F, the capacitance change appears to be >10 times in size compared to the example shown in Figure 1A. If this is not a simple typo, an explanation is required as to whether this finding even applies to the smaller events.

We would like to clarify that this is not a typo. It is reasonable from our observations that endocytic events with bigger size are typically associated with bigger ionic current drifts. For demonstration purpose, an endocytic event with relatively big size was, therefore, chosen in Figure 1F. To answer the reviewer’s question of whether the ionic current flowing through the patch applies to smaller events, we have compared ionic current drifts with different capacitance sizes (such as < 1 fF vs > 1 fF). To make sure that each group has enough endocytic events for this comparison, we grouped all endocytic events only at positive membrane potentials by neglecting negative membrane potential due to the outwardly rectifying properties of this current drift as shown in Figure 1G. The ionic current drifts were also divided by the voltage across the patch membrane to minimize the impact from the difference in patch membrane voltages. Our results showed similar normalized amplitudes of this endocytosis-associated current drift between relatively smaller (< 1 fF) and bigger (> 1 fF) endocytic events, suggesting that this finding may apply to all endocytic events regardless of their size. This result is added as Figure 1—figure supplement 3 in the revised manuscript.

5. The authors aimed to study the physiological function of presynaptic TRPM7, and they conclude that TRPM7 might act as a Ca^2+^-influx pathway. However, the pipette-solutions for whole-cell experiments (in chromaffin cells and HEK-cells, where TRPM7-related currents were apparently detected directly, in Figure 1F and Figure 4A) did not contain Mg^2+^ (in chromaffin cells Na-ATP and Na-GTP were used; in HEK-cells no nucleotides or magnesium were included), and TRPM7 is described as a ion channel that is fully inhibited by 2 mM intracellular Mg-ATP (Nadler et al., 2001). Therefore, it is not straight forward to see how Ca^2+^ would permeate the channel under physiological conditions. The authors have data (e.g. CGaMP6) consistent with Ca^2+^-influx, and they show that a mutation (LCF) blocking permeation also blocks function in chromaffin cells, but this is not conclusive evidence that permeation is involved. The authors need to discuss the negative regulation of TRPM7 by Mg-ATP carefully (it seems to not even be mentioned), and whether physiologically relevant conditions are likely to arise where this block would be relieved. They should also discuss their choice of pipette solutions. Finally they should discuss whether their data might be consistent with other scenarios than Ca^2+^-influx through TRPM7.

As pointed out by the reviewer, during electrophysiological recordings, TRPM7 can be fully inhibited by intracellular Mg^2+^ (Nadler et al., 2001), which justifies our choice of whole-cell pipette solutions with no Mg^2+^ for electrophysiological recordings in both chromaffin cells and HEK 293 cells. However, roles of influx of ions, including Ca^2+^, via TRPM7 in neurons has been documented under both physiological (Krapivinsky et al., 2006) and pathological (Aarts et al., 2003; Sun et al., 2009) conditions, indicating that additional cytoplasmic signaling molecules, which remains unknown so far, may be able to facilitate TRPM7 to bypass the tonic inhibitions from intracellular Mg^2+^ within intact neurons. In addition, TRPM7may be blocked by extracellular Mg^2+^ (Nadler et al., 2001), and this Mg^2+^-dependent blockage of TRPM7 can be relieved upon membrane depolarizations (Kerschbaum et al., 2003; Li et al., 2007), a mechanism similar to postsynaptic NMDA receptors (Hunt and Castillo, 2012). Therefore, it is likely that membrane depolarizations will enhance Ca^2+^ influx via TRPM7 and thus accelerate synaptic vesicle endocytosis. Interestingly, many neurons, such as cortical neurons, frequently generate bursts of action potentials in vivo (Baranyi et al., 1993; Gray and McCormick, 1996), and these bursts can occur at a very high frequency, up to 300 Hz, in awake animals as reported (Gray and Viana Di Prisco, 1997). Since synaptic vesicle endocytosis becomes a limiting factor for synaptic transmission in neurons during high frequency action potential firings (Kawasaki et al., 2000), it is likely that neurons may require an up-regulation of synaptic vesicle endocytosis to maintain synaptic transmission. Therefore, it is speculated that roles of TRPM7 in synaptic vesicle endocytosis will be enhanced during high frequency firings, which may couple synaptic vesicle endocytosis to neuronal activities. This discussion has been added to “implications” of Discussion section in the revised manuscript.

Following the reviewer’s comment regarding whether our data might be consistent with other scenarios that Ca^2+^ influx through TRPM7, we have also discussed the possibility of TRPM7’s kinase domain in synaptic vesicle endocytosis: While our data, by showing the non-conducting TRPM7^LCF^ mutant fails to rescue endocytic defects in both TRPM7 KO chromaffin cells and neurons, indicates TRPM7 as an ion channel in synaptic vesicle endocytosis. Alternatively, it could be possible that this might reflect roles of TRPM7’s C-terminal kinase in synaptic vesicle endocytosis, since TRPM7’s kinase activity could be potentially altered in the non-conducting TRPM7 mutant. However, this possibility is less likely, since annexin A1 (Dorovkov et al., 2011), the so far identified substrate of TRPM7 kinase important for vesicle endocytosis (Futter and White, 2007), is largely expressed in ependymal and glial cells rather than neurons in the brain (Solito et al., 2008).

This discussion has been added in the revised manuscript. We appreciate the reviewer’s comments, which promote us to discuss the physiological relevance of Mg^2+^-mediated inhibition on TRPM7 in synaptic vesicle endocytosis.

6. It is unclear why the KO-induced changes in vesicle fusion do not show-up as a slower rise in the SypHy measurements shown in Figure 3. This should be briefly discussed.

As presented in Figure 6A-B in inhibitory synapses and in Figure 6C-E in excitatory neurons, there is an enhanced short-term synaptic depression upon repetitive stimulations in TRPM7 KO neurons. However, as pointed out by the reviewer, this KO-induced enhanced synaptic depression does not show up as a slower increase in the SypHy measurement as shown in Figure 3A and C, which, we believe, may be due to a couple of reasons: (1) in order to estimate synaptic vesicle endocytosis, the traces presented in Figure 3A and C is normalized to the peak of fluorescent signals upon sustained stimulation, which can cover up any potential changes in pHluorin signal increase during stimulations in KO neurons; and (2) as reported previously, there is substantial amount of synaptic vesicle endocytosis during sustained stimulations and the fluorescent signal increase during stimulations is not a good representation of synaptic vesicle exocytosis but rather represents the net balance of exocytosis an endocytosis (Sankaranarayanan and Ryan, 2001).

7. The explanatory connection that the authors make between the TRPM7-KO effects on calcium signals, endocytosis, and short-term plasticity seems forced. The KO reduces presynaptic calcium signals (likely also with endocytosis during stimulation trains) and increases synaptic depression already very early in the train. There are more parsimonious explanations for the short-term plasticity change: Reduced bulk calcium levels could cause reduced calcium-dependent activation of the vesicle priming machinery or even reduced activation of calcium sensors in fusion (Synaptotagmins). This aspect should be considered in the discussion part of the paper. It may well be that the short-term plasticity defect in the TRPM7 KO neurons is not directly related to the endocytosis defect. Further, it seems hard to imagine that a selective slow-down of endocytosis would become manifest as an appreciable change in exocytosis already after only 10 APs – unless one considers 'site clearing' as proposed by Neher and Sakaba.

We agree with the reviewer that an enhanced short-term depression in KO neurons may possibly be due to alterations in exocytosis and release site clearance, in addition to synaptic vesicle endocytosis as we have proposed. As pointed by the reviewer, since it is postulated that clearance of endocytic components from release sites is an important factor for shortterm depression (Hua et al., 2013; Neher, 2010), the enhanced short-term depression we have observed in TRPM7 KO neurons may indicate a potential role of TRPM7 in release site clearance, in addition to synaptic vesicle endocytosis are we have proposed. It is also shown that the degree of short-term depression can be modulated by exocytic parameters, such as recruitments of release-ready vesicles (Neher and Sakaba, 2008). A reduction in presynaptic Ca^2+^ signals during sustained activity in TRPM7 KO neurons (Figure 5C-D) could slow down recruitments of release-ready vesicles, by reducing activations of vesicle priming and/or Ca^2+^ sensors in fusion (Chapman, 2008; Pang and Sudhof, 2010), which could also lead to an enhanced short-term depression. Following the reviewer’s comments, this has been added as part of “potential roles of TRPM7 in short-term synaptic depression” in the Discussion section in the revised manuscript.

8. It is unclear why the 'endocytosis-associated conductance' the authors describe was missed in the many studies employing whole-cell recordings in chromaffin cells. Do the authors assume that detection failed in these cases due to the outward-rectifying character and/or the fact that typical measurements are done at -80 mV? This issue should be discussed.

We agree with the reviewer on this comment. Previous whole-cell recordings in chromaffin cells are typically conducted at its resting membrane potential of around -80 mV (Smith and Neher, 1997), a condition that may have missed the detection of TRPM7 currents due to its outward-rectifying properties (Nadler et al., 2001). Additionally, most of previous whole-cell recordings in chromaffin cells includes 0.5-1 mM Mg^2+^ in the intracellular solution, which substantially inhibits TRPM7 currents (Nadler et al., 2001). This is another big reason for the missed detection of TRPM7 currents by previous studies in chromaffin cells. This discussion has been added in the revised manuscript.

9. The time scale of changes in endocytosis kinetics in TRPM7 KO chromaffin granules is of no known physiological impact. This should be discussed.

Vesicle fission is a crucial step during endocytosis (Kaksonen and Roux, 2018). The importance of vesicle fission in endocytosis can be reflected by significant roles of the GTPase dynamin, a key molecule essential for vesicle fission (Antonny et al., 2016), in synaptic vesicle endocytosis (Ferguson et al., 2007; Newton et al., 2006). Our previous work using cell-attached capacitance recordings, the same approach as the present study, has shown that inhibition of dynamin GTPase slows down endocytic kinetics of single vesicles in chromaffin cells at the millisecond time scale. Therefore, our data shows an increase in the endocytic kinetics of single vesicle in KO chromaffin cells, indicating a potential role of TRPM7 in vesicle fission during endocytosis. The physiological significance of TRPM7 in vesicle fission during endocytosis is reiterated by a slower rate of synaptic vesicle endocytosis in TRPM7 KO neurons. The endocytic rate of synaptic vesicles is decided by the number of endocytic vesicles and the kinetics of single endocytic events. Our results, showing slower kinetics (Figure 1A and E) but no change in the number of single endocytic events (Figure 1B) in TRPM7 KO chromaffin cells, indicate that TRPM7 may regulate the rate of synaptic vesicle endocytosis by regulating the kinetics of single endocytic events but not the number of endocytic vesicles. Following the reviewer’s comment, this discussion has been added to the Discussion section in the revised manuscript.

Essential – Requiring Additional Experiments1. A basic analysis of the morphology of TRPM7 KO chromaffin cells and neurons is needed to complement the present data. Further, it should be tested if the TRPM7 KO affects the expression or localization of proteins that are critically involved in presynapse function, especially in endocytosis and endocytotic fission (e.g. Dynamins).

Following the reviewer’s comments, we have analyzed, using immunostaining, expression levels of endocytic proteins, such as dynamin 1, AP2, clathrin and synaptotagmin 1, in both chromaffin cells and presynaptic terminals from TRPM7 KO animals. Chromaffin cells were identified with immunostaining of tyrosine hydroxylase (TH) and presynaptic terminals were detected with a synaptophysin antibody, and the fluorescent signals of individual endocytic proteins were normalized to TH signals in chromaffin cells and synaptophysin signals in presynaptic terminals in neurons to minimize variations from experiments to experiments. This analysis showed no alterations in expression levels of all these 4 endocytic proteins in both chromaffin cells and presynaptic terminals from TRPM7 KO neurons, suggesting that the endocytic defects we have observed is unlikely due to any changes in these key endocytic proteins. These additional data have been added as the Figure 3—figure supplement 3 for chromaffin cells and the Figure 1—figure supplement 4 for presynaptic terminals in neurons in the revised manuscript.

2. It should be properly verified that WT and mutant versions of TRPM7 are expressed and targeted in the same fashion. Figure S5 does not support the case that the mutant protein is correctly trafficked. It is unclear what the anti-FLAG staining corresponds to in the KO cells infected with a virus with an empty vector. This staining appears to be vesicular, which is also present in the TRPM7-WT infected cells, but not in the LCF infection. A higher resolution approach is required to localize the FLAG staining (confocal might be sufficient). If TRPM7 is no longer targeted correctly when it has the conduction mutation, one cannot conclude that the current through the channel is what is important.

As shown in the original supplementary Figure 5A (now Figure 4—figure supplement 2A in the revised manuscript), since KO cells in the mock group are infected with a lentivirus with an empty vector, the anti-FLAG signals detected in this group is likely non-specific staining. We agree with the reviewer that a higher resolution approach may be better to localize the FLAG staining. Following the reviewer’s comment, we have taken confocal images of KO chromaffin cells infected with either WT or mutant TRPM7, showing similar distributions of these 2 TRPM7 versions within KO cells. These confocal images are now added as panel C in this Figure 4—figure supplement 2 in the revised manuscript.

3. Given the ability of the authors to perform very high-end electrophysiological analyses of transmitter secretion from chromaffin cells, the use of KCl^-^depolarization to assess TRPM7-KO-induced changes in exocytosis from chromaffin cells appears 'crude'. This type of stimulation is massive and somewhat unphysiological. The key question that remains open is whether exocytosis triggered by more physiological stimuli is affected by the KO (e.g. as assessed by short depolarisation stimuli combined with capacitance measurements). This should be tested in a small KO-vs.WT analysis.

Following the reviewer’s comment, we have compared exocytosis induced by membrane depolarization between WT and TRPM7 KO chromaffin cells using whole-cell capacitance measurements. Our results showed no difference in exocytosis induced by membrane depolarization, suggesting no change in exocytosis in TRPM7 KO chromaffin cells. This result is consistent with our amperometrical data using KCl^-^depolarization as shown in Figure 2, and the data from whole-cell capacitance recordings is now added as the Figure 2—figure supplement 1 in the revised manuscript.

4. Figure 3E-F show reacidification in WT and KO neurons. The description of this experiment is insufficient. The rationale of the experiment is a bit diffuse and it is unclear how the quantification (i.e. on what part of the trace). Moreover, the trace in panel E seems to be inconsistent with the quantification in panel F. In panel E, the reduction in DeltaF in the presence of the second low-pH pulse is steeper in the WT than in the KO. Can this be expllained? In essence, these data in their present state are not sufficient to draw unequivocal conclusions regarding reacidification. The authors should include controls with bafilomycin, to understand whether the slopes they measure are actually reporting reacidification.

As pointed out by the reviewer, our data on vesicle reacidification seems confusing, which, we think, could be due to low signal-to-noise ratio of some of our original data on individual synaptic terminals. Therefore, we have performed additional experiments between WT and TRPM7 KO neurons. We believe the data from these new experiments has excellent signal-to noise ratio for individual synaptic terminals, which is evident from the example shown in Figure 3—figure supplement 2. We believe the high quality of new data set will allow us to draw an unequivocal conclusion on roles of TRPM7 in vesicle reacidification. Results from these new experiments show no difference in the averaged trace (Figure 3E in the revised manuscript) and time constant of vesicle acidification (Figure 3F in the revised manuscript) between WT and TRPM7 KO neurons. The revised manuscript has been updated with results from these new data.

Following the reviewer’s comment, we also performed additional experiments with bafilomycin to verify whether the decay we measured are reporting vesicle reacidification. Our data shows that the decay can be blocked by applications of bafilomycin that inhibits vesicular proton pump, demonstrating that the decay we measured does represent vesicle reacidification. A sample trace of an individual terminal in the presence of bafilomycin is presented in the Figure 3—figure supplement 2.

In response to the reviewer’s concern of “insufficient description on vesicle reacidification”. We have now added, in the legends of this Figure 3—figure supplement 2, the following description on our experiments on vesicle reacidification in much more details:

To measure reacidification rate of newly formed endocytic vesicles using sypHy as described by Atluri and Ryan (J. Neurosci., 2006), neurons infected with lentivirus encoding sypHy were rapidly perfused with a valve controlled pressurized local perfusion system (ALA Scientific, Farmingdale, NY). Surface fluorescence of pHluorin is quenched by the first application of acidic buffer (pH 5.5 buffer with the HEPES substituted with equi-molar MES) for 10s to establish the baseline, and the quenched fluorescence is recovered by washing off the acidic solution. To isolate pHluorin signals from newly endocytosed synaptic vesicles, acidic buffer, indicated by the 2^nd^ bar in the trace, is applied with a 20s duration 5s after the cessation of stimulation train. Newly endocytosed alkaline pool of pHluorin is detached from the plasma membrane and thus resistant to quenching by the applied acidic buffer, and the decay of this portion of fluorescence to the baseline represents acidification of newly endocytosed vesicles. The decay of fluorescent signals, which can be blocked by bafilomycin A1, thus likely reflects reacidification kinetics of newly endocytic vesicles.

Reacidification rate of newly endocytosed vesicles is estimated by an exponential fit, as indicated by the red dashed line, of this fluorescent signal decay from the first to the last point during this 20s duration of acidic buffer application.

We believe these descriptions, in combined with the traces added in the Figure 3—figure supplement 2, will be sufficient and convincing for our experiments on vesicle reacidification.

5. Reacidification kinetics are not resolved, yet are reported as a single parameter of an exponential fit. Furthermore, when the acid was removed, the KO cells should have had higher fluorescence, since it is decaying more slowly (e.g. difference WT vs. KO at the 80 s time point). The acid quenching experiment showed no difference. Given that the authors are not resolving an exponential decay of the acid-quenched surface signal, it would be more appropriate to simply report what fraction of the signal early on after the end of the stimulus – where there is a real difference in the amount of the signal that has decayed (e.g. at the 60 s time point) – can be quenched by acid perfusion. If it is the same fraction for WT and KO, the differences are not explained by the build-up of an alkaline pool, which is really what the authors want to assess here.

As mentioned in our response to reviewer’s previous comment on vesicle reacidification, the data from our new experiment, which has an excellent signal-to-noise ratio as shown in the Figure 3—figure supplement 2, clearly shows that the reacidification kinetics can be reported as a single exponential fit in our hands, which is consistent with previous reports (Atluri and Ryan, 2006; Egashira et al., 2015). Additionally, our data showed that, upon the removal of acidic buffer at the 80 s time point, presynaptic terminals from KO neurons had a higher fluorescence (WT: 0. 221 ± 0.011; KO: 0.287 ± 0.012; p < 0.01, unpaired two-tailed student’s t-test), as shown in Figure 3E, which is in line with the reviewer’s prediction. This higher fluorescence in KO neurons reflects a build-up of surface signals during applications of acidic buffer after stimulations, thus suggesting an endocytic defect in TRPM7 KO neurons.

Alternatively, the reviewer indicates to report whether there is a difference in the fraction of the signal early on after the end of the stimulus – where there is a real difference in the amount of the signal that has decayed (e.g. at the 60 s time point) – can be quenched by acid perfusion. However, as shown in Figure 3 A, C and E, the endocytic defect in TRPM7 KO neurons builds up gradually from the cessation of stimulations to the end of recordings and there is no obvious difference in fluorescent signals within 8 s after the end of stimulation. Since our acid perfusion is applied 5s right after the end of stimulation, synaptic vesicle endocytosis during this 5s duration, from the end of stimulations to the start of acid perfusion, is relatively small. We think our method may not be sensitive enough to catch a potentially very small difference in synaptic vesicle endocytosis, occurring during the 5s duration (from the end of stimulations to the start of acid perfusion), between WT and TRPM7 KO neurons.

We appreciate the reviewers’ comments on vesicle reacidification, which prompt us to perform additional experiments, thus making our conclusion on vesicle reacidification more convincing.

6. It is important to know if the phenomena described in the present study occur under physiological conditions (i.e. 37{degree sign}C, ~1.25 mM Ca). The temperature in particular is important as cooling can substantially distort the relevance of different molecular processes. The reviewers do not expect a repetition of all experiments at 37{degree sign}C. The easiest might be an experiment on pHluorin assessment of vesicle recycling in neurons, showing that at 37{degree sign}C there is a measurable impact on the pHluorin recovery kinetics due to the absence or presence of external Ca^2+^.

To answer the reviewer’s question, we have performed a side-by-side comparisons of synaptic vesicle endocytosis between WT and TRPM7 KO neurons at 34*ºC* in the presence of 2 mM extracellular Ca^2+^. Our results showed that synaptic vesicle endocytosis is much slower in TRPM7 KO neurons as compared to WT neurons at physiological temperature (p < 0.001), which is consistent with our data observed at room temperature (Figure 3). This result is now added as Figure 3—figure supplement 1 in the revised manuscript. We thank the reviewer for this comment, which strengths our conclusion on the role of TRPM7 in synaptic vesicle endocytosis.

References:

Aarts, M., Iihara, K., Wei, W.L., Xiong, Z.G., Arundine, M., Cerwinski, W., MacDonald, J.F., and Tymianski, M. (2003). A key role for TRPM7 channels in anoxic neuronal death. Cell 115, 863-877.

Akerboom, J., Chen, T.W., Wardill, T.J., Tian, L., Marvin, J.S., Mutlu, S., Calderon, N.C., Esposti, F., Borghuis, B.G., Sun, X.R., et al. (2012). Optimization of a GCaMP calcium indicator for neural activity imaging. J Neurosci 32, 1381913840.

Antonny, B., Burd, C., De Camilli, P., Chen, E., Daumke, O., Faelber, K., Ford, M., Frolov, V.A., Frost, A., Hinshaw, J.E., et al. (2016). Membrane fission by dynamin: what we know and what we need to know. EMBO J 35, 2270-2284. Atluri, P.P., and Ryan, T.A. (2006). The kinetics of synaptic vesicle reacidification at hippocampal nerve terminals. J Neurosci 26, 2313-2320.

Baranyi, A., Szente, M.B., and Woody, C.D. (1993). Electrophysiological characterization of different types of neurons recorded in vivo in the motor cortex of the cat. II. Membrane parameters, action potentials, current-induced voltage responses and electrotonic structures. Journal of neurophysiology 69, 1865-1879.

Brockhaus, J., Bruggen, B., and Missler, M. (2019). Imaging and Analysis of Presynaptic Calcium Influx in Cultured Neurons Using synGCaMP6f. Front Synaptic Neurosci 11, 12.

Chapman, E.R. (2008). How does synaptotagmin trigger neurotransmitter release? Annu Rev Biochem 77, 615-641.

Dorovkov, M.V., Kostyukova, A.S., and Ryazanov, A.G. (2011). Phosphorylation of annexin A1 by TRPM7 kinase: a switch regulating the induction of an α-helix. Biochemistry 50, 2187-2193.

Egashira, Y., Takase, M., and Takamori, S. (2015). Monitoring of vacuolar-type H^+^ ATPase-mediated proton influx into synaptic vesicles. J Neurosci 35, 3701-3710.

Ferguson, S.M., Brasnjo, G., Hayashi, M., Wolfel, M., Collesi, C., Giovedi, S., Raimondi, A., Gong, L.W., Ariel, P., Paradise, S., et al. (2007). A selective activity-dependent requirement for dynamin 1 in synaptic vesicle endocytosis. Science 316, 570-574.

Futter, C.E., and White, I.J. (2007). Annexins and endocytosis. Traffic 8, 951-958.

Gray, C.M., and McCormick, D.A. (1996). Chattering cells: superficial pyramidal neurons contributing to the generation of synchronous oscillations in the visual cortex. Science 274, 109-113.

Gray, C.M., and Viana Di Prisco, G. (1997). Stimulus-dependent neuronal oscillations and local synchronization in striate cortex of the alert cat. The Journal of neuroscience : the official journal of the Society for Neuroscience 17, 32393253.

Hua, Y., Woehler, A., Kahms, M., Haucke, V., Neher, E., and Klingauf, J. (2013). Blocking endocytosis enhances shortterm synaptic depression under conditions of normal availability of vesicles. Neuron 80, 343-349.

Hunt, D.L., and Castillo, P.E. (2012). Synaptic plasticity of NMDA receptors: mechanisms and functional implications. Current opinion in neurobiology 22, 496-508.

Jin, J., Desai, B.N., Navarro, B., Donovan, A., Andrews, N.C., and Clapham, D.E. (2008). Deletion of Trpm7 disrupts embryonic development and thymopoiesis without altering Mg^2+^ homeostasis. Science 322, 756-760.

Kaksonen, M., and Roux, A. (2018). Mechanisms of clathrin-mediated endocytosis. Nat Rev Mol Cell Biol 19, 313-326.

Kawasaki, F., Hazen, M., and Ordway, R.W. (2000). Fast synaptic fatigue in shibire mutants reveals a rapid requirement for dynamin in synaptic vesicle membrane trafficking. Nature neuroscience 3, 859-860.

Kerschbaum, H.H., Kozak, J.A., and Cahalan, M.D. (2003). Polyvalent cations as permeant probes of MIC and TRPM7 pores. Biophysical journal 84, 2293-2305.

Krapivinsky, G., Mochida, S., Krapivinsky, L., Cibulsky, S.M., and Clapham, D.E. (2006). The TRPM7 ion channel functions in cholinergic synaptic vesicles and affects transmitter release. Neuron 52, 485-496.

Li, M., Du, J., Jiang, J., Ratzan, W., Su, L.T., Runnels, L.W., and Yue, L. (2007). Molecular determinants of Mg^2+^ and Ca^2+^ permeability and pH sensitivity in TRPM6 and TRPM7. The Journal of biological chemistry 282, 25817-25830.

Nadler, M.J., Hermosura, M.C., Inabe, K., Perraud, A.L., Zhu, Q., Stokes, A.J., Kurosaki, T., Kinet, J.P., Penner, R., Scharenberg, A.M., et al. (2001). LTRPC7 is a Mg.ATP-regulated divalent cation channel required for cell viability. Nature 411, 590-595.

Neher, E. (2010). What is Rate-Limiting during Sustained Synaptic Activity: Vesicle Supply or the Availability of Release Sites. Front Synaptic Neurosci 2, 144.

Neher, E., and Sakaba, T. (2008). Multiple roles of calcium ions in the regulation of neurotransmitter release. Neuron 59, 861-872.

Newton, A.J., Kirchhausen, T., and Murthy, V.N. (2006). Inhibition of dynamin completely blocks compensatory synaptic vesicle endocytosis. Proc Natl Acad Sci U S A 103, 17955-17960.

Pang, Z.P., and Sudhof, T.C. (2010). Cell biology of Ca^2+^-triggered exocytosis. Curr Opin Cell Biol 22, 496-505.

Saheki, Y., and De Camilli, P. (2012). Synaptic vesicle endocytosis. Cold Spring Harb Perspect Biol 4, a005645.

Sankaranarayanan, S., and Ryan, T.A. (2001). Calcium accelerates endocytosis of vSNAREs at hippocampal synapses. Nat Neurosci 4, 129-136.

Sgobio, C., Kupferschmidt, D.A., Cui, G., Sun, L., Li, Z., Cai, H., and Lovinger, D.M. (2014). Optogenetic measurement of presynaptic calcium transients using conditional genetically encoded calcium indicator expression in dopaminergic neurons. PLoS One 9, e111749.

Singh, M., Lujan, B., and Renden, R. (2018). Presynaptic GCaMP expression decreases vesicle release probability at the calyx of Held. Synapse 72, e22040.

Smith, C., and Neher, E. (1997). Multiple forms of endocytosis in bovine adrenal chromaffin cells. J Cell Biol 139, 885894.

Solito, E., McArthur, S., Christian, H., Gavins, F., Buckingham, J.C., and Gillies, G.E. (2008). Annexin A1 in the brain-undiscovered roles? Trends Pharmacol Sci 29, 135-142.

Sun, H.S., Jackson, M.F., Martin, L.J., Jansen, K., Teves, L., Cui, H., Kiyonaka, S., Mori, Y., Jones, M., Forder, J.P., et al. (2009). Suppression of hippocampal TRPM7 protein prevents delayed neuronal death in brain ischemia. Nat Neurosci 12, 1300-1307.

Yao, L.H., Rao, Y., Varga, K., Wang, C.Y., Xiao, P., Lindau, M., and Gong, L.W. (2012). Synaptotagmin 1 is necessary for the Ca^2+^ dependence of clathrin-mediated endocytosis. J Neurosci 32, 3778-3785.

[Editors' note: further revisions were suggested prior to acceptance, as described below.]

Your manuscript has been very substantially improved. However, the reviewers have identified three issues that need to be resolved before the paper can be published in eLife:1. The new confocal images added to Figure 3 (supplement 3, panels A-D; supplement 4, panels A, C, E, G) and to Figure 4 (supplement 2, panel C) are of very poor quality (very low resolution, blurred, pixelated) and do not look like confocal images (rather like ones that have been improperly digitally compressed). In fact, the new panel C in Figure 4, supplement 2 appears to be of even less resolution than the original images in panel A. In essence, the problematic panels indicated above are not of publication quality and need to be redone.

We appreciate reviewer’s comment. We realized that the confocal images mentioned by the reviewers were captured with pinhole of 200 µm, which gave a relatively poor quality of images. To obtain confocal images with better quality, we have re-captured the confocal images with pinhole set to 50 µm. Consequently, all the confocal images related to Figure 3 (supplement 3, panels A-D; supplement 4, panels A, C, E, G) and to Figure 4 (supplement 2, panel C) have been replaced with our recaptured images with much higher quality.

2. The description of confocal microscopy in the methods part is insufficient and requires more detail (i.e. numerical aperture and type of objective, light source, pinhole setting, etc.).

Following the reviewer’s comment, we have added more description on confocal imaging in “Materials and methods”. Regarding methods on confocal imaging, it reads as “Confocal fluorescence images of chromaffin cells or neurons were captured using an Olympus FluoviewFV10i laser-scanning confocal microscope with a 60x oil-immersion objective (NA 1.4) and 473nm, 559nm, and 635nm lasers. The pinhole was set to 50 μm for imaging capturing, and the image size was 1024 x 1024.” now in the revised manuscript.

3. Regarding Figure 5, supplement 1, it seems that the 340 nm images in panel A are saturated. If so, it would be impossible to do a ratio analysis as shown in panel B. This issue needs to be explained and resolved.

We appreciate that the reviewer made a valid point, and we agree that any potential saturation will for sure affect data analysis. However, we believe that none of our images used for analysis were saturated. Our Photometrics Prime 95B sCMOS camera saturates when the intensity has reached 65000 a.u. Author response image 1 shows intensity plots for the raw images that were used for panel A of Figure 5, supplement 1, and the intensity seems far below 65000 a.u of our camera’s saturating limit. In the meantime, we realized that the brightness of those images was adjusted for visualization. To minimize any potential concerns on saturation of displayed images, we have replaced with new images that have reduced brightness but exactly same display settings between WT and KO cells in the revised manuscript.

**Author response image 1. sa2fig1:**